



# Total Column Water Vapour Retrieval from S-5P/TROPOMI in the Visible Blue Spectral Range

Christian Borger, Steffen Beirle, Steffen Dörner, Holger Sihler, and Thomas Wagner

Satellite Remote Sensing Group, Max Planck Institute for Chemistry, Mainz, Germany

**Correspondence:** C. Borger (christian.borger@mpic.de) and T. Wagner (thomas.wagner@mpic.de)

**Abstract.** Total column water vapour has been retrieved from TROPOMI measurements in the visible blue spectral range and compared to a variety of different reference data sets for clear-sky conditions during boreal summer and winter. The retrieval consists of the common two-step DOAS approach: first the spectral analysis is performed within a linearized scheme and then the retrieved slant column densities are converted to vertical columns using an iterative scheme for the water vapour a priori

profile shape which is based on an empirical parameterization of the water vapour scale height. Moreover, a modified albedo map was used combining the OMI LER albedo and scaled MODIS albedo map. The use of the alternative albedo is especially important over regions with very low albedo and high probability of clouds like the Amazon region.

The errors of the TCWV retrieval have been theoretically estimated considering the contribution of a variety of different uncertainty sources. For observations during clear-sky conditions, over ocean surface, and at low solar zenith angles the error

typically is around values of 10-20% and during cloudy-sky conditions, over land surface, and at high solar zenith angles it reaches values around 20-50%.

In the framework of a validation study the retrieval demonstrates that it can well capture the global water vapour distribution: the retrieved $H_2O$ VCDs show very good agreement to the reference data sets over ocean for boreal summer and winter whereby the modified albedo map substantially improves the retrieval's consistency to the reference data sets in particular over tropical

landmasses. However over land the retrieval underestimates the VCD by about 10%, particularly during summertime. Our investigations show that this underestimation is likely caused by uncertainties within the surface albedo and the cloud input data: Low level clouds cause an underestimation but for mid to high level clouds good agreement is found. In addition, our investigations indicate that these biases can probably be further reduced by the use of updated cloud input data.

The TCWV retrieval can be easily applied to further satellite sensors (e.g. GOME-2 or OMI) for creating uniform measurement

data sets on longterm which is particularly interesting for climate and trend studies of water vapour.

## 1 Introduction

Water vapour is the most important natural greenhouse gas in the atmosphere and plays a key role in the atmospheric energy balance via radiative effects and latent heat transport (Held and Soden, 2000). Due to its high spatiotemporal variability on all atmospheric scales, accurate knowledge of the amount and distribution of water vapour is essential for numerical weather

prediction and climate monitoring.





Several in situ and remote sensing measurement techniques have been developed in the past decades, enabling to observe the water vapour distribution from platforms like radiosondes, balloons, aircrafts and satellites. The particular absorption properties of water vapour allow to retrieve the water vapour content via satellites for several different spectral ranges from the radio (Kursinski et al., 1997), microwave, e.g. AMSU (Rosenkranz, 2001), thermal infrared, e.g. AIRS (Susskind et al., 2003), near-

infrared, e.g. MODIS (Gao and Kaufman, 2003) and MERIS (Bennartz and Fischer, 2001) to the visible, e.g. GOME (Noël et al., 1999; Wagner et al., 2003; Lang et al., 2007), SCIAMACHY (Noël et al., 2004), and GOME-2 (Grossi et al., 2015).

In the visible spectral range total column water vapour (TCWV) has so far been retrieved mostly in the "red" spectral range because the absorption is strongest there. However for this spectral range the ocean surface albedo is relatively low leading to a low sensitivity for the lowermost troposphere where the highest water vapour concentrations occur. In addition current

and past satellite sensors can not resolve the fine absorption structure of water vapour in this spectral range causing non-linear absorption effects (e.g. saturation) which have to be accounted for in post-processing. Thus Wagner et al. (2013) suggested to apply retrievals in the "blue" spectral range (around 442 nm) where the absorption is much weaker than in the red making the retrieval problem quasi-linear. In addition the ocean surface albedo is much higher leading to a higher sensitivity of the near-surface layers. First operational analyses of a similar approach have been performed by Wang et al. (2019) for measurements

of the Ozone Monitoring Instrument (OMI, Levelt et al., 2006).

In October 2017 the TROPOspheric Monitoring Instrument (TROPOMI, Veefkind et al., 2012) onboard ESA's Sentinel-5 Precursor (S-5P) satellite was launched in a sun-synchronous polar orbit with an equator crossing time of 13:30 local time. TROPOMI is a UV-Vis-NIR push-broom spectrometer and consists of 450 detectors/rows covering a swath width of 2600 km. The outstanding property of TROPOMI is that its spectral bands in the visible combine a high signal to noise with an un-

precedented spatial resolution of $3.5 \times 7.5 \, \text{km}^2$ (and $3.5 \times 5.6 \, \text{km}^2$ since August 2019) at nadir which allows to perform spectral analyses at a never seen before accuracy even on small spatial scale.

In this paper we introduce a TCWV retrieval based on the spectral analysis approach of Wagner et al. (2013) to S-5P/TROPOMI observations. The paper is organized as follows: In Sect. 2 we give an overview of the retrieval describing general retrieval principles and presenting the retrieval set-up. In Sect. 3 we present an empirical parameterization of the a priori water vapour

profile shape and an iterative scheme making use of the relation between the water vapour profile shape and TCWV. In Sect. 4 we evaluate different input albedo products and in Sect. 5 we perform a detailed uncertainty analysis including a variety of different error sources. In Sect. 6 we present first TCWV results retrieved from TROPOMI measurements and perform a validation study using data sets from satellite, ground-based measurements, and reanalysis models as reference. In Sect. 7 we draw conclusions and summarize the outcomes of our investigations.

## 2  Retrieval principles

### 2.1  Wavelength calibration and spectral analysis

In a first step the wavelength alignment of the measured irradiance is calibrated for each of the 450 TROPOMI detectors/rows via a nonlinear least-squares fit in intensity space using the solar spectrum from Kurucz (1984) as reference. Simultaneously


the instrumental spectral response function (ISRF) is approximated assuming an asymmetric Super-Gaussian following the
definition of Beirle et al. (2017):

$$
S_{asym}(x) = \begin{cases} \exp\left(-\left|\frac{x}{w-a_w}\right|^k\right) & \text{for x} \leq 0 \\ \exp\left(-\left|\frac{x}{w+a_w}\right|^k\right) & \text{for x} > 0 \end{cases} \tag{1}
$$

Next, we perform a spectral analysis using the differential optical absorption spectroscopy (DOAS; Platt and Stutz, 2008)
scheme in which the attenuation along the light path is calculated via Beer-Lambert's law in optical depth space:

$$
\ln\left(\frac{I}{I_0}\right) = \tau \approx -\sum_i \sigma_i(\lambda) \cdot \text{SCD}_i + \Phi \tag{2}
$$

where $i$ denotes the index of a trace gas of interest, $\sigma_i(\lambda)$ its respective absorption cross section, $\text{SCD}_i = \int_s c_i ds$ its concentration integrated along the light path $s$ (the so called slant column density), and $\Phi$ a closure polynomial accounting for Mie and Rayleigh scattering as well as low-frequency contributions.

Table 1 summarizes the fit setup of the retrieval's spectral analysis. The retrieval's fit window ranges from 430 nm to 450 nm and accounts for molecular absorption by water vapour (HITRAN 2008, Rothman et al., 2009), $NO_2$ at 220K (Vandaele et al., 1998), ozone (Serdyuchenko et al., 2014) and the $O_2$-$O_2$ dimer (Thalman and Volkamer, 2013). In order to account for the Ring effect we include two Ring spectra (Wagner et al., 2009) and for $\Phi$ we use a 5th order polynomial. Furthermore we include pseudo-absorbers accounting for intensity offset, for shift and stretch effects (Beirle et al., 2013) and, for ISRF changes along the orbit (Beirle et al., 2017) for the ISRF parameters $w$ and $k$ in Eq. (1). All molecular absorption cross sections are convolved with the ISRF of the corresponding TROPOMI row/detector determined during the calibration process.

The molecular absorption by water vapour within our fit window is relatively weak and hence the modelled line lists vary strongly from HITRAN 2008 to HITRAN 2012 (Rothman et al., 2013) and to HITRAN 2016 (Gordon et al., 2017). Thus the choice of line list is afflicted by a high degree of uncertainty. Lampel et al. (2015) found out that HITRAN 2012 underestimates the water vapour concentration derived from Long Path DOAS observations by approximately 10% and that the previous version HITRAN 2008 agrees better to the reference measurements. Further Long Path DOAS measurements taken during the CINDI-2 campaign also confirm the findings from Lampel et al. (2015) (see Appendix B for more details). Thus combining the findings from Lampel et al. (2015) and Wang et al. (2019) we conclude that HITRAN 2008 fits best our needs and is superior to the most recent version of the HITRAN line lists (HITRAN 2016).

Due to the high daily data volume of the TROPOMI L1B radiances (about 40 gigabyte per day for Band 4), the execution of a non-linear fit without high performance infrastructure is demanding in computation time. Therefore, we implemented a weighted linear least squares fit for our retrieval, in which the weights are the fractional coverage of the pixel within the fit window (details in Appendix A). This weighting of the outermost pixels of the fit window avoids "jumps" of pixels included in the DOAS fit, as it would occur for a fixed fit window due to the changing pixel-to-wavelength mapping across track. Thus, across track "stripes" in the SCDs are avoided.

Figure 1 illustrates a typical example of such a spectral analysis of a TROPOMI measurement spectrum in which the absorption





structures of water vapour, $NO_2$, and the Ring effect can be well identified and the residual spectrum showing a mainly noisy structure. Figure 2 depicts the distribution of the $H_2O$ SCD from one TROPOMI orbit (orbit number 6930) on 13th February 2019. It demonstrates that the TROPOMI retrieval is able to capture the meso- to macro-scale water vapour patterns like convective updrafts in the tropics and atmospheric rivers in the midlatitudes whereby the small $H_2O$ SCD values in the tropics are caused by cloud shielding.

## 2.2    VCD conversion and Box-AMF simulations

To convert the slant column density to a vertical column density (VCD), we apply the so called airmass factor (AMF):

$$\text{VCD} = \frac{\text{SCD}}{\text{AMF}}$$

The airmass factor accounts for the non-trivial effects of the atmospheric radiative transfer and is usually based on radiative transfer model (RTM) simulations. In our case we used the 3D Monte Carlo RTM McArtim (Deutschmann et al., 2011) and

performed simulations at a wavelength of $442\,\text{nm}$ for different retrieval scenarios (summarized in Tab. 2) assuming an aerosol-free atmosphere. These simulations yield a Jacobian vector $J = \frac{\partial \ln I}{\partial \beta}$ defined at each grid box $i$ and the altitude-dependent AMFs (BAMF) can be calculated according to the formula:

$$\text{BAMF}_i = -\frac{J_i}{I\Delta h}$$

with the simulated intensity $I$ normalised by the solar spectrum and the box thickness $\Delta h$. These BAMF profiles have to be

combined with the partial vertical columns $\text{VCD}_i$ of an a priori water vapour profile:

$$\text{AMF} = \frac{\sum_i \text{BAMF}_i \cdot \text{VCD}_i}{\sum_i \text{VCD}_i} \tag{3}$$

For the case of a cloud contaminated pixel we assume that the cloud is a Lambertian reflector with an albedo of 80% and use the cloud top height as surface altitude input for the AMF. Under the assumption of the independent pixel approximation the resulting cloud-affected AMF can then be calculated as a linear combination of the AMF for a clear-sky scenario and the AMF

for a cloudy-sky scenario weighted by the respective simulated intensities $I$ and the effective cloud fractions $\zeta$ as follows:

$$\text{AMF} = \frac{(1-\zeta)I_{clear}\text{AMF}_{clear} + \zeta \cdot I_{cloud}\text{AMF}_{cloud}}{(1-\zeta)I_{clear} + \zeta \cdot I_{cloud}}$$

## 3    A priori water vapour profile shape

As described in Sect. 2.2 and Eq. (3), knowledge of the a priori water vapour profile shape is necessary for accurate calculations of the AMF from BAMF profile. However, simply assuming the same a priori profile shape for the whole globe might cause

biases because it can not account for the atmospheric variability of water vapour, such as latitudinal variation, seasonal cycles, or different profile shapes over maritime and continental regions due to different water vapour source (e.g. evapotranspiration by plants). Also simply using profiles from numerical weather models is not uncritical: for instance Wang et al. (2019) found





out that their calculated AMF change strongly depends on which reanalysis model data they were using.

Weaver and Ramanathan (1995) approximated the water vapour profile by an exponential decay with altitude and a correspond-
ing scale height defined as

$$H_{vapor} = \frac{R_v \langle T \rangle^2}{L \langle \Gamma \rangle}$$

where $\langle T \rangle$ denotes the mean air temperature within an atmospheric column, $\langle \Gamma \rangle$ the mean lapse rate within the same atmo-
spheric column, $R_v$ the gas constant of water vapour, and $L$ the specific latent heat. However this definition requires knowledge
of the mean air temperature and/or the lapse rate and that the relative humidity is constant with altitude. The former can be
only estimated using numerical weather models and the latter is very unlikely to occur in the atmosphere.

Thus we investigate to find an empirical parameterization of the scale height and thereby focus on its dependency on the $H_2O$
VCD and the aforementioned atmospheric variabilities, i.e. dependencies of latitude, seasonal cycle and surface properties
(such as vegetation effects).

For these investigations we use profile data retrieved from measurements of the Constellation Observing System for Meteorol-
ogy, Ionosphere, and Climate (COSMIC, Anthes et al., 2008) program provided by ROMSAF. The COSMIC data are based
on the GPS radio occultation (RO) technique which provides high resolution vertical profiles of bending angles (Hajj et al.,
2002) that can be used to retrieve the atmospheric refractivity $N$. Since the atmospheric refractivity $N$ is given by (Smith and
Weintraub, 1953):

$$N = 77.6 \frac{p}{T} + 3.73 \times 10^5 \frac{E}{T^2} \tag{4}$$

with the air pressure $p$, the air temperature $T$, and the water vapor pressure $E$, GPS RO allows to retrieve profile information
under all-weather conditions with a high vertical resolution of approximately $100\,\mathrm{m}$ in the lower troposphere up to $1\,\mathrm{km}$
in the stratosphere (Anthes, 2011) and an accuracy of around $1\,\mathrm{g/kg}$ (Heise et al., 2006; Ho et al., 2010b) while having an
almost uniform global distribution (Ho et al., 2010a). We use data retrieved between 2013 and 2016, which accumulates to
approximately $1.6 \times 10^6$ profiles.

## 3.1   Calculation of scale height

For the calculation of the scale height we highsample the COSMIC profile to a $100\,\mathrm{m}$ grid up to $14\,\mathrm{km}$ or rather only consider
profile data below $150\,\mathrm{hPa}$ (close to the tropopause height). Then we sum up all the partial columns of the COSMIC profile
data from ground up to a (scale-) height $H_{sum}$ where 63% of the $H_2O$ VCD are reached:

$$\frac{\int_0^{H_{sum}} n(z)dz}{\int_0^{TOA} n(z)dz} > 1 - \frac{1}{e} \approx 63\%$$

In order to evaluate this scale height approach we performed a synthetic study in which we compared AMFs calculated for the
original COSMIC water vapour profile measurements with AMFs for an exponential profile with the corresponding calculated
scale height $H_{sum}$. For the simulation of the BAMF profiles we assume an albedo of 7%. The solar zenith angle is calculated





for the location of the COSMIC profile assuming an hour angle of 90° and the line of sight angles are prescribed for −90°, −70° and −50°.

The results of the intercomparison are given in Figure 3. The 2D histograms reveal that the AMFs derived with the exponential profile agree well with the AMFs calculated directly from the COSMIC profiles indicating that the chosen method can well reproduce the shapes of the COSMIC profiles. This good agreement can be also observed in the histograms of Fig. 4 which illustrate distributions of relative deviation between the AMFs for for selected latitude bins. These distributions have a sharpe shape and peak around values of 0% indicating that the AMFs from the exponential shape are almost unbiased to the reference

AMFs. In addition Fig. 5 shows examplary profiles for cases of good and bad agreement to the reference AMFs for the same selected latitude bins as in Figure 4. In general the bad agreement (left column) occurs for profile shapes in which a distinctively strong gradient is observed in the lower troposphere and from that quasi-constant values with altitude. Nevertheless the maximal absolute relative AMF-deviations only have values around 15%.

The results of the intercomparison for prescribed cloudy-sky conditions and nadir viewing geometry are illustrated in Fig. 6

in which the panels show histograms of the relative AMF deviation for the same selected latitude bins as in Fig. 4 but for different cloud fraction (10%, 20% and 50%; left to right column) and cloud top height (1 km, 2 km and 5 km; top to bottom row) scenarios. For a cloud top height of 1 km the AMFs calculated from the exponential profiles are generally biased negative for all cloud fractions, in particular for the latitude bin of -30° to -20°N. However for higher clouds the AMFs agree well with the reference AMFs for almost all cloud scenarios except the extreme case with a cloud fraction of 50% and a cloud top height

of 5 km or more.

Alternative methods for calculating the scale height yielded systematic overestimations of the AMF for clear-sky conditions (Fig. A7) and higher scatter within the AMF for cloudy-sky conditions (Fig. A8) in comparison to the sum method, as shown in detail in Appendix C.

### 3.2 Parameterization of scale height

Figure 7 depicts the distribution of the calculated COSMIC scale height $H_{sum}$ against the COSMIC TCWV for boreal summer over ocean for latitude bins of 10°. The regression fits (solid red lines) are based on orthogonal distance regression (ODR) using the "scipy.odr" package built on ODRPACK (Boggs et al., 1992). For low latitudes (tropics and subtropics) the scale height shows a high linear correlation to the $H_2O$ VCD with slopes around 0.04 and Pearson correlation coefficients $R$ of 70% and above. In contrast for high latitudes the slope increases up to 0.1 and also the scatter increases distinctively, i.e. the

correlation coefficient only reaches values of around 0.3 in the polar regions. This decrease in linear agreement is likely caused by the higher atmospheric variability due to higher atmospheric dynamics in the midlatitudes. Also the uncertainty is higher in COSMIC profile because a drier atmosphere leads to a smaller sensitivity of the COSMIC profile retrieval to water vapour concentrations (compare Eq. (4) and Kursinski et al. 1997).

Figure 8 illustrates the same panels as Fig. 7 but for data over land. In general the scatter for all latutude bins has increased

distinctively resulting in an inferior linear agreement between the $H_2O$ VCD and the scale height compared to the data over ocean, especially for deserts and northern polar regions. Fortunately, the surface albedo of these regions is usually high and





thus the AMF is less dependent on the a priori profile shape. In addition these regions are governed by an arid climate and thus the retrieved $H_2O$ VCDs are expected to be small. Correspondingly the absolute $H_2O$ VCD errors due to uncertainties in the AMF are still relatively small.

In the following we investigate a parameterization of the scale height with respect to $H_2O$ VCD, latitude, season for ocean and land separately.

### 3.2.1   Ocean

The regression line parameters of the ODR fit results between COSMIC TCWV and COSMIC scale height for each latitude bin for each month for data over ocean are illustrated in Figure 9. The values for the fitted slopes (left panel in Fig. 9) indicate

a quadratic dependency with latitude and reveal a seasonal shift towards higher latitudes during July, August, and September. Also the values for the fitted intercept vary with latitude and season.

Thus the scale height over ocean $H_{ocean}$ can be approximated as follows:

$$H_{ocean}(\text{VCD}, \theta, t) = \alpha(\theta, t) \cdot \text{VCD} + \beta(\theta, t) \tag{5}$$

with

$$\alpha(\theta, t) = a_0(t) + a_1(t) \cdot \theta + a_2(t) \cdot \theta^2$$
$$\beta(\theta, t) = b_0(t) + b_1(t) \cdot \theta + b_2(t) \cdot |\theta - \theta_0(t)| \tag{6}$$

with the latitude $\theta$ and the day of year $t$. The annual variation of the function parameters $a_i$, $b_i$ and $\theta_0$ from Eq. (6) fitted for the monthly data sets (illustrated in Fig. 9) is depicted in Figure A9. Most function parameters reveal an annual and semi-annual cycle over the year. Hence these function parameters can be approximated by a superposition of two simple cosine functions

with prescribed frequencies:

$$a_i(t) = a_{i1} \cdot \cos(a_{i2} + \omega t) + a_{i3} \cdot \cos(a_{i4} + 2\omega t) + a_{i5}$$

with $t$ as the day of year and $\omega = \frac{2\pi}{365}$. Such functions have also been fitted and illustrated for the monthly data in Fig. A9 (solid orange lines) whereby we assumed that the day of year representing the month is the first day of the month. For most function parameters the fits coincide well with the data points and in the cases of suboptimal fit results the annual variation of the data

is relatively small, indicating that our choice of parameterization is valid.

Altogether we have to fit 35 parameters to the complete data set of calculated COSMIC scale heights for the parameterization of the scale height over ocean. The goodness of the parameterization in approximating the scale height is illustrated in Fig.10 for different latitude zones. For the latitude zones including the tropics (-15° to 15°N) and subtropics (-35° to 35°N) we find a good agreement between the parameterization and the calculated COSMIC scale height with $R^2$ of 0.72 and 0.60 respectively.

However including higher latitudes in the evaluation, i.e. midlatitudes (-60°to 60°N) and polar regions (-90°to 90°N), leads to an increased scatter and a worsening of the parameterization ($R^2$ of 0.45 and 0.44 respectively). This inferior agreement is likely caused by the larger atmospheric variability in the midlatitudes (e.g. higher atmospheric dynamics) as well as an increased uncertainty in the COSMIC water vapour profile measurements due to lower water vapour concentrations.



### 3.2.2 Land

Figure 8 already revealed much larger scatter in the distribution of COSMIC TCWV and COSMIC scale height for data over land indicating that the water vapour profile shape over land surface is less homogeneous than over ocean likely due to further heterogenously distributed water vapour sources such as evapotranspiration by plants and soil. Thus the $H_2O$ VCD and scale height are likely to be dependent on the amount of vegetation, i.e. high vegetation is associated with high evapotranspiration and high water vapour concentrations near the ground and thus the scale height should be close to the scale height over ocean. In

contrast to that low amounts of vegetation are associated with less evapotranspiration and a usually drier atmosphere indicating that the scale height should be higher than over ocean.

For quantifying the amount of vegetation we use the Normalized Difference Vegetation Index (NDVI) where a value of 1.0 indicates high vegetation and a value around 0.0 indicates low vegetation. As data source for the NDVI we use data within the MODIS Aqua MYDC13C2 Version 6 product (Didan et al., 2015) and continue as follows: First we calculate the parameterized

scale height $H_{ocean}$ assuming an ocean surface globally. Then we calculate the ratio of the calculated COSMIC scale height over land $H_{land}$ and the parameterised scale height $H_{ocean}$.

Figure 11 shows the ratio $H_{land}/H_{ocean}$ as a function of the NDVI for data sets filtered by different landcover types and the solid red lines represent the robust regression results (summarized in Tab. 3) using the model from Siegel (1982). The left panel depicts the distribution for which no filter is applied. Except for low NDVI values a linear relation between ratio and

NDVI is observable, however for NDVI values around 0.1 the ratio varies strongly between 0.7 and 3.0. In the center panel we use the landcover classification from the MODIS Aqua MCD12C1 Version 6 product (Sulla-Menashe et al., 2019) to filter measurements for locations classified as landcover type 15 (corresponding to a desert). With this filter the ratio now only varies between 0.7 and 1.5 with a weak dependence on the NDVI. If we further filter locations of landcover type 7 (corresponding to open shrublands) the fit results of the robust regression change only slightly compared to the first filtered data set.

Hence the scale height over land $H_{land}$ can be approximated as the scale height over land $H_{ocean}$ multiplied by a first order polynomial of the NDVI:

$$H_{land} = H_{ocean}(\text{VCD}, \theta, t) \cdot (\gamma_{land} + \delta_{land} \cdot \text{NDVI}) \tag{7}$$

whereby in the following we use the results for the data set filtered for landcover type 7 and 15 globally. Since regions of landcover type 7 or 15 are usually arid, the retrieved $H_2O$ VCD is small and thus the error due to an inadequate parameterization

of the AMF is much smaller than the fit error of the spectral analysis.

### 3.3 Iterative retrieval scheme

For the calculation of the $H_2O$ VCD we precomputed AMF look-up tables (LUT) for the different water vapour profile shapes with scale heights ranging from 0.5 km to 5.0 km. These LUTs can then be used within a fixed-point iteration. In our case the iterative retrieval scheme is based on a fixed-point iteration according to Steffensen's method (Steffensen, 1933; Wendland and





Steinbach, 2015):

$$\mathrm{VCD}_{i+1} = \mathrm{VCD}_i - \frac{(f(\mathrm{VCD}_i) - \mathrm{VCD}_i)^2}{f(f(\mathrm{VCD}_i)) - 2 \cdot f(\mathrm{VCD}_i) + \mathrm{VCD}_i}$$

where $f$ is a function calculating the scale height for a given VCD using Eq. (5) and (7), applying it to the precomputed AMF look-up tables and from that returning a new VCD. The advantage of Steffensen's method is that it does not need a derivative and is able to determine the fixed-point even for the case of a non-contractive function (Wendland and Steinbach, 2015). For the first guess we derived the initial VCD from the SCD using a geometric AMF ($\mathrm{AMF}_{\mathrm{geo}} = \frac{1}{\cos(\mathrm{SZA})} + \frac{1}{\cos(\mathrm{VZA})}$) and stop the iteration as soon as the logarithmic difference between two consecutive results is smaller than 5% (approximately $1\,\mathrm{kg\,m^{-2}}$ assuming an average $H_2O$ VCD of $20\,\mathrm{kg\,m^{-2}}$) or after six iteration steps. We also checked other values for the first guess and could confirm that the convergence of the iterative scheme is independent from them.

Figure 12 illustrates a comparison of $H_2O$ VCD distributions for the cases of using a global constant a priori water vapour profile shape (left panel) with a scale height of $2\,\mathrm{km}$ (in accordance to Weaver and Ramanathan (1995)) and using the iterative scale height approach (center panel) for all-sky conditions (i.e. no cloud filter applied) during an atmospheric river event at the Western coast of the US on 13th February 2019. The right panel of Fig. 12 depicts the distribution of the water vapour scale height yielded during the iterative VCD conversion. The water vapour scale height varies a lot along the orbit and differs distinctively from $2\,\mathrm{km}$ causing large deviations between the two approaches particularly at pixels with high TCWV values and for clouded pixels. Yet in contrast to the approach with a constant scale height the iterative approach is still able to give reasonable TCWV results and does not exceed values higher than $80\,\mathrm{kg\,m^{-2}}$.

Similiar results can be found in Fig. 13 which illustrates the $H_2O$ VCD distributions of both approaches for the same scenario for clear-sky (effective cloud fraction CF<20%, top row) and all-sky (CF≤100%, bottom row) conditions. In addition Fig. 13 depicts the TCWV distribution from microwave sensor SSMIS f16 which has a temporal difference of around +2.3 hours. For the clear-sky case the VCD distributions between both approaches are almost identical whereby for the constant scale height approach (left panel) very high VCDs (exceeding values higher than $80\,\mathrm{kg\,m^{-2}}$) can be observed at the edges of the cloudy regions in the Northern subtropics. For the all-sky case (bottom row) the differences between the two approaches are largest in cloudy regions, but as already mentioned before even under these unfavourable observation conditions the iterative approach is still able to give reasonable VCD values, whereas the constant scale height approach distinctively overestimates the VCD. In addition the iterative approach shows an overall good agreement to the SSMIS observations.

## 4 Evaluation of different surface albedo input data

The surface albedo has a strong impact on the radiative transfer and thus also on the AMF. Hence we investigated the impact of different albedo products on the TCWV retrieval: the OMI monthly a) mean and b) minimum Lambertian equivalent reflectance (LER) at $442\,\mathrm{nm}$ from Kleipool et al. (2008) and c) MODIS Aqua blue surface reflectance from the MODIS MYD13C2 Version 6 product (Didan et al., 2015). The MODIS reflectance covers a broad spectral window from $459\,\mathrm{nm}$ to $479\,\mathrm{nm}$. Thus to account for the different spectral windows of the albedo products we scale the MODIS albedo by factor of 0.9. This factor





was estimated by calculating the ratio between 472 nm/442 nm of the OMI yearly minimum LER over parts of Australia where cloud contamination is generally low and hence the OMI LER has reasonably accurate values.

Figure 14 illustrates the global mean $H_2O$ VCD of boreal summer 2018 for the different albedo input data over land (top row: monthly mean OMI LER, middle row: monthly minimum OMI LER, bottom row: scaled monthly MODIS Aqua blue surface reflectance). In the tropical and subtropical regions the OMI albedos cause a distinctive separation of the VCDs between land and ocean, in particular at the coasts of South America, Africa and Indonesia. These aforementioned regions are often affected by cloud cover, which might cause that the OMI albedo statistics can not filter cloudy cases correctly, so that cloud-contaminated observations are used within the albedo calculations. As a consequence the values in the OMI albedo are too high and lead to an overestimation of the AMF which in turn causes an underestimation of the $H_2O$ VCD.

In contrast to that MODIS pixels have a much higher spatial resolution and MODIS' NIR channels are more sensitive to cloud contimination, yielding a higher sample size and allowing for correct cloud filtering. Hence the $H_2O$ VCD distribution using the MODIS surface reflectance results in a much smoother transition from ocean to land and in general much higher VCD values over land along the equator. Thus in the following we use a combination of the MODIS and OMI albedos: the scaled MODIS Aqua blue surface reflectance over land and the monthly minimum OMI albedo over ocean.

## 5 Uncertainty estimation

The error budget of the $H_2O$ VCD is determined by the propagation of the main error sources of the fitted SCD and the precalculated AMF. Errors in the SCD are mainly caused by random errors like the photon noise, and systematic errors, e.g. the uncertainty of the absorption cross section whereas errors in the AMF are mostly systematic with random contributions.

### 5.1 Uncertainties in the slant column density

Table 4 summarizes the different error sources for the $H_2O$ SCD and the corresponding estimated uncertainties. As demonstrated in Sect. 2.1 and Appendix B the water vapour absorption cross section varies distinctively between the different HITRAN versions. Hence we assumed that the uncertainty of the water vapour cross section is of the same order of magnitude as the changes between the cross section versions, i.e. approximately around 10%.

The retrieval's spectral analysis directly yields the $1\sigma$ standard fit error of the $H_2O$ SCD which is usually dominated by noise. For a better understanding of these fit errors we separated them into data for small/large solar zenith angles (SZA<20° and 70°<SZA≤90°, respectively), low/high surface albedo (<3% and >15%, respectively), and clear-/cloudy-sky observation conditions (CF<5% and CF>20%, respectively). The distributions of the standard and relative fit errors of the spectral analysis are given in Fig. 15 and 16, respectively. The median values in Fig. 15 indicate that the standard errors for high SZA (around $0.3 \times 10^{23}$ molec cm$^{-2}$) are twice as high as for small SZA (around $0.15 \times 10^{23}$ molec cm$^{-2}$). Under clear-sky conditions the standard error for small surface albedo values is larger than for high surface albedo, but for cloudy conditions it does not depend on the surface albedo.

Figure 16 reveals that the relative fit errors for high SZA are higher than for low SZAs. However the locations of maximal prob-





ability density and the medians also indicate that the distributions are positive skewed in particular for high SZA scenarios:

for these scenarios the relative errors easily exceed values of 100%. Nevertheless using the locations of maximal probability density as a rule-of-thumb estimate, relative fit errors have values around 10% for low SZAs and approximately 30% for high SZAs.

## 5.2   Uncertainties in the AMF

The uncertainty in the AMF depends on the uncertainty of its input parameters. Because the parameters of the viewing ge-

ometry (i.e. solar zenith angle, line of sight angle, and solar relative azimuth angle) are known with high accuracy the most important uncertainties are uncertainties of the surface albedo, cloud fraction, cloud top height, and water vapour profile shape. In order to estimate the contribution of each input parameter to the overall AMF uncertainty we define standard scenarios (summarized in Tab. 5) for which we calculate the AMF from the precalculated LUT and then vary the input parameter for each scenario according to its uncertainty assumption listed in Table 6. The uncertainties of the water vapour scale height have

been derived from the fit results of the intercomparisons between the measured COSMIC scale height and the parameterized scale height over ocean (see Fig. 10) and land (see Fig. 11).

Figure 17 depicts box-whisker plots of the relative AMF error due to uncertainties in surface albedo and scale height for the standard clear-sky scenarios of surface albedo, solar zenith angle, and scale height. It reveals that uncertainties in surface albedo and scale height over low vegetation have strongest impact on the AMF and can cause AMF errors larger than 30%, in

particular for scenarios with low surface albedo or high solar zenith angle. On average the median values of the AMF errors typically vary around approximately 10%.

Figure 18 illustrates box-whisker plots of the relative AMF error due to uncertainties in surface albedo, scale height, cloud fraction. and cloud top height for all standard scenarios listed in Table 5. In contrast to the clear-sky scenarios the impact of the surface albedo uncertainties has strongly decreased, but in general the contributions of all AMF errors have increased

distinctively. The main source for the AMF errors is still the uncertainty of the scale height over low vegetation whose median values varies between 20-50%, but can also cause AMF errors larger than 60%.

Table 6 summarizes the results of the different error sources considered in the AMF uncertainty for clear- and cloudy-sky conditions. For clear-sky conditions one can typically assume a relative AMF error around 10-15% and for cloudy-sky conditions around 10-25%.

## 335   5.3   Total $H_2O$ VCD uncertainty

The total relative $H_2O$ VCD uncertainty can be approximated by

$$\frac{\Delta \mathrm{VCD}}{\mathrm{VCD}} = \sqrt{\left(\frac{\Delta \mathrm{AMF}}{\mathrm{AMF}}\right)^2 + \left(\frac{\Delta \mathrm{SCD}}{\mathrm{SCD}}\right)^2} \tag{8}$$

With our findings of typical relative AMF and $H_2O$ SCD uncertainties the total relative VCD uncertainty is typically around 10-20% for observations during clear-sky conditions, over ocean surface, and at low solar zenith angles. During partly clouded-sky

conditions, over land surface, and at high solar zenith angles the error reaches values of approximately 20-50%.





## 6   Validation study

In order to evaluate the retrieval's performance we conducted a validation study for the time ranges of boreal summer (June,
July, and August) 2018 and boreal winter (December, January, February) 2018/2019 whereby we only include clear-sky ob-
servations (i.e. pixels with a effective cloud fraction smaller than 20%) and ice- and snowfree pixels. As reference data for the

validation we use TCWV from the Special Sensor Microwave Imager/Sounder (SSMIS), from the reanalysis model ERA-5
and from the ground-based GPS network SuomiNet.

As cloud input data we use the cloud information (effective cloud fraction at 440 nm and cloud top height) as well as the surface
altitude from the TROPOMI L2 $NO_2$ product (Van Geffen et al., 2019) and as surface albedo input data we use the combination
of the modified MODIS and OMI albedo described in Section 4.

### 6.1   SSMIS comparison

For the evaluation we use measurements from SSMIS onboard NOAA's f16 and f17 satellite processed by Remote Sensing
Systems (RSS) and provided by NASA Global Hydrology Resource Center on a daily 0.25°×0.25° grid. SSMIS can observe
the TCWV distribution under all-sky conditions over ocean with an accuracy of around $1\,kg\,m^{-2}$ (Wentz, 1997; Mears et al.,
2015). Since SSMIS changes its equator crossing time (ECT) we only include SSMIS observations whose ECT is within 3

hours (and 5 hours for f17, respectively) with respect to TROPOMI's ECT of 13:30 LT. For the intercomparison we only
include SSMIS measurements that are not affected by rain.

Figure 19 depicts the comparison between SSMIS (f16, top row and f17, bottom row) and TROPOMI for boreal summer
(left column) and winter (right column). For f16 (top row) the scatter is distributed closely along the 1-to-1 diagonal (dashed
lines) for both seasons and the fitted regression lines (red solid lines) indicate a very good agreement between both data with

slopes around 0.96, intercepts around $-1.6\,kg\,m^{-2}$ for summer and $-1.7\,kg\,m^{-2}$ for winter and coefficients of determination
of $R^2 = 0.91$. For f17 the comparison reveals similar agreement with slopes around 0.97 and intercepts around $-1.5\,kg\,m^{-2}$
with $R^2 = 0.89$ for both seasons. Overall considering the differences in collocation time (3 hours and 5 hours for f16 and f17,
respectively) the comparison shows that the TROPOMI TCWV retrieval can well capture the water vapour distribution over
ocean.

To investigate the influence of clouds on our retrieval, we plot the difference (top row) and relative difference (bottom row)
between TROPOMI and SSMIS as a function of the input cloud top height (CTH) in Figure 20 and 21 for f16 and f17,
respectively. The median over the whole CTH range (blue dashed line) indicates an underestimation of the TROPOMI $H_2O$
VCD of approximately 12-13% ($2.6\,kg\,m^{-2}$). However the large majority of data points is distributed within the CTH bin
between 0-1 km revealing that the underestimation of the TROPOMI TCWV is mainly caused by low clouds. For mid clouds

the median difference almost cancels out whereas for high clouds it first increases and then remains almost constant with cloud
top height.

Further validation results for SSMIS f16 and f17 separated into different cloud fraction and cloud top height bins for July
2018 are given in Fig.A10 and Fig. A11 respectively. The results indicate that there is no dependency with cloud fraction but a





distinctive dependency with cloud top height: The retrieval underestimates for clouds below 1 km, is in very good agreement

for mid level clouds (1 km to 4 km) and overestimates for higher clouds.

## 6.2  ERA-5 comparison

For the intercomparison between the reanalysis model ERA-5 and TROPOMI we use ERA-5 TCWV data provided by Copernicus Climate Change Service (2017) on a 0.25°× 0.25° grid. We only take into account values which are within 1 hour with respect starting sensing time of the TROPOMI orbit and separate the data into data over ocean and data over land.

The results of the intercomparison are summarized in Figure 22. Over ocean (top row in Fig. 22) the results are similar to the results from the comparison between TROPOMI and SSMIS: Apart from slopes close to 0.95 and intercepts close to zero, the linear regression yields $R^2$ of 94% for summer and 95% for winter, respectively. Over land the linear regression still yields high values of the coefficient of determination $R^2$, but the TROPOMI retrieval generally underestimates the $H_2O$ VCD by approximately 12% during summer (and 7% during winter). Since the values of the correlation coefficient are still high and the

values over ocean coincide very well with the reference data sets, we assume that this underestimation has to be caused by a systematic uncertainty within the input parameters for our retrieval.

The influence of the cloud top height input is illustrated in Fig. 23 for data over ocean. The median is around $-1.6\,\mathrm{kg\,m^{-2}}$ ($-7.1\%$) and $-1.3\,\mathrm{kg\,m^{-2}}$ ($-6.7\%$) during summer and winter, respectively, whereby similar to SSMIS these underestimations are caused by the majority of data points within the 0-1 km CTH bin. For increasing CTH the deviation from the reference

increases and leads to an overestimation. For data over land (Fig. 24) the CTH variability is much larger than over ocean, i.e. most data points are now distributed between 0 to 3 km and the median is around values of $-1.5\,\mathrm{kg\,m^{-2}}$ ($-10.3\%$) and $-0.4\,\mathrm{kg\,m^{-2}}$ ($-4.0\%$) during summer and winter respectively. Furthermore low clouds still cause an underestimation and for mid to high clouds the deviations almost cancel out, but one can also observe an increasing scatter for winter data.

All these findings reveal that the combination of albedo uncertainties and uncertainties in the cloud properties (cloud frac-

tion and cloud top height) as well as in the scale height parameterization have a distinctive influence on the AMF. The cloud products from TROPOMI rely on the OMI albedo which, as we have demonstrated in Sect. 4, has several problems over land surface. In addition the uncertainty of the OMI albedo over land surface is higher than over ocean due to a highly spatiotemporal variability of the scenery and the differences between the monthly minimum and the monthly mean albedo are higher over land than over ocean. Furthermore the cloud top height is calculated via the cloud top pressure and has to be combined

with the surface pressure. Thus the uncertainty of the cloud top height over land is higher than over ocean since over ocean the topography is much simpler.

Nevertheless the complex radiative interactions between albedo and clouds might amplify or cancel out these deviations and thus make it difficult to draw clear conclusions.

As for the SSMIS comparison, further validation results for ERA-5 over ocean and land separated into different cloud fraction

and cloud top height bins for July 2018 are given in Fig. A12 and Figure A13. Similar to SSMIS, the results over ocean reveal an underestimation for low clouds and an overestimation for high clouds that there is almost no dependency with cloud fraction.





Over land low clouds still cause an underestimation, however for cloud top heights above 2 km the retrieval shows very good agreement to ERA-5 indicating that the input cloud top height for our retrieval is too low.

### 6.3 SuomiNet/GPS comparison

For the intercomparison with TCWV from ground-based GPS we use data from the network SuomiNet (Ware et al., 2000) provided by UCAR. SuomiNet stations are distributed over North and Central America and provide data every 30 min with a typical accuracy of $2 \, \mathrm{kg \, m^{-2}}$ (Duan et al., 1996; Fang et al., 1998). Thus we only take into account TROPOMI pixels within a distance of 0.1° to the GPS station and within 2 hours with respect to the GPS measurement.

Figure 25 illustrates scatter plots of the intercomparison between TROPOMI and SuomiNet for boreal summer and winter. For
both seasons the robust regression indicates an underestimation of around 20% (i.e. slopes of 0.82 and 0.84) with high Pearson correlation coefficients of 88%. In order to investigate the influence of clouds on our retrieval, we plot the difference (top row) and the relative difference (bottom row) between TROPOMI and Suominet as a function of the input cloud top height (CTH) in Figure 26. The median over the whole CTH range (blue dashed line) indicates an underestimation of the TROPOMI $H_2O$ VCD of approximately 14% ($3.5 \, \mathrm{kg \, m^{-2}}$) during summer and of 8% ($0.8 \, \mathrm{kg \, m^{-2}}$) during winter. However during summer the
median values for each 1 km CTH bin (blue dots) reveal that the underestimation is mainly caused by low clouds whereas for mid and high clouds the median difference almost cancels out. During winter this pattern is not clearly observable due to much larger scatter but also here low clouds mainly cause the underestimation in TCWV whereby the difference is generally within the range of accuracy of the SuomiNet retrieval.

Figure A14 depicts further validation results separated into different cloud fraction and cloud top height bins for boreal summer
2018. Though the sample size is much smaller, similar results as for SSMIS and ERA-5 are obtained: indenpendent from the cloud fraction low clouds cause an underestimation of around 15-20% whereas for mid clouds the TROPOMI $H_2O$ VCDs show much better agreement to the SuomiNet TCWV and for high clouds TROPOMI overestimates by around 10%.

### 7 Summary and conclusions

In this paper, we introduce a total column water vapour retrieval from TROPOMI spectra in the visible blue spectral range
using an iterative vertical column conversion scheme and provide a detailed characterization of our retrieved $H_2O$ VCD by performing a detailed uncertainty analysis and intercomparisons to reference data sets from the microwave sensor SSMIS, from the reanalysis model ERA-5 and from the ground-based measurements GPS network SuomiNet.

For the iteration scheme we describe the a priori water vapour profile as an exponential decay with a scale height $H$ and developed an empirical parameterization for this scale height. This parameterization is based on COSMIC water vapour profile
data and relates the a priori water vapour profile shape to the $H_2O$ VCD, the seasonal cycle, the latitude and the vegetation (and NDVI, respectively). We demonstrate that we can correctly reproduce the scale heights in particular for data at low latitudes (tropics and subtropics). However we also observe an increasing scatter if higher latitudes are included in the comparison, likely because of the higher variability in $H_2O$ VCD due to midlatitudinal cyclone dynamics and a general higher uncertainty





in the COSMIC profile data for drier atmospheric conditions. Overall, the retrieved profile heights are very reasonable and we obtain a substantial improvement using the new parameterisation compared to the use of a prescribed constant water vapour profile.

For the uncertainty analysis we investigated the impact of several error sources on the $H_2O$ SCD and AMF like clouds, surface albedo, profile shape and instrument properties. The error estimation reveals that the main SCD uncertainty is the fit error of the spectral analysis and that the main AMF uncertainties are caused by uncertainties in the surface albedo and water vapour profile shape. For the $H_2O$ VCD we estimated a typical total relative error of around 10-20% for observations during clear-sky conditions, over ocean surface, and at low solar zenith angles. For observations during cloudy-sky conditions, over land surface, and high solar zenith angles the error reaches values of approximately 20-50%. Thus the theoretically estimated errors are of the same order of magnitude as the deviations found during the retrieval's evaluation. However uncertainties in the absorption cross section of water vapour are a further systematic error source that can additionally contribute up to 10%. Based on the LP-DOAS comparisons we estimate these errors to be around 5% for this study, so that they are negligible compared to the other error sources.

In the validation study we demonstrate that for clear-sky conditions the retrieved TROPOMI $H_2O$ VCDs over ocean are in very good agreement to the reference data sets and can correctly capture the global water vapour distribution. Over land the TROPOMI retrieval can reproduce the TCWV distribution however we also observe a distinctive underestimation of around 10% in particular during boreal summer.

Nevertheless these underestimations might be caused by the uncertainties of the external input data for the retrieval: For instance the OMI LERs from Kleipool et al. (2008) are too high over tropical landmasses likely due to incorrect cloud filtering which causes too high AMFs leading to too low $H_2O$ VCDs. Although we tried to overcome this issue by using a surface reflectance product from MODIS Aqua, the cloud products from the TROPOMI L2 $NO_2$ product still rely on the OMI LER for calculating the effective cloud fraction and cloud top height and thus also have a large uncertainty. The intercomparisons to the reference data sets show that these uncertainties in the cloud products have a substantial impact on the $H_2O$ VCD: Our investigations reveal that the input cloud top height is probably too low which in turn leads to higher AMFs and consequently to an underestimation in TCWV. Yet one has to consider that the radiative properties of the cloud and albedo products interact at a high degree of complexity so that a clear explanation or suggestion on how to overcome these issues is beyond the scope of this paper.

Overall the successful application of the TCWV retrieval in the visible blue spectral range on TROPOMI measurement is very promising for further investigations including application to further satellite sensors such as OMI, SCIAMACHY, and GOME-1/2 or the upcoming Sentinel-4 instrument and expanding the retrieval to measurements contaminated by higher cloud fractions. As the retrieval allows for a fast execution of large data sets investigations of longterm trends using a TCWV data set of merged timeseries of different satellite sensors are easily possible. However, since these data sets have to be uniform they require consistent input data across the different satellite sensors, in particular for cloud products.



*Data availability.* The TROPOMI TCWV data presented here are available upon request.



## Appendix A: Weighted linear least squares fit for spectral analysis

To handle the daily high data volume of TROPOMI and to avoid "jumps" of pixels included in the fit window we implemented
a weighted linear least squares fit for the DOAS analysis. The weights $W$ are the fractional coverage of the pixel within the fit
window (see also Fig. A1):

$$W(\lambda) = \begin{cases} 1 - \frac{|\lambda - \lambda_{low}|}{\Delta\lambda} & \frac{|\lambda - \lambda_{low}|}{\Delta\lambda} < 1 \land \lambda - \lambda_{low} < 0 \\ 1 & \lambda_{low} < \lambda < \lambda_{up} \\ 1 - \frac{|\lambda - \lambda_{up}|}{\Delta\lambda} & \frac{|\lambda - \lambda_{up}|}{\Delta\lambda} < 1 \land \lambda - \lambda_{up} > 0 \\ 0 & \text{else} \end{cases}$$

with $\lambda_{low}$ and $\lambda_{up}$ the lower and upper boundary of the fit window and $\Delta\lambda$ the average wavelength increment within the fit
window. The elements of the weight matrix are then given as $w_{ii} = \sqrt{W_{ii}(\lambda_i)}$. Hence Eq. (2) can be solved by simple linear
algebra:

$$\mathbf{y}' = \mathbf{M}'\mathbf{x}$$

$$\hat{\mathbf{x}} = \left(\mathbf{M}'^{\mathbf{T}}\mathbf{M}'\right)^{-1}\mathbf{M}'^{\mathbf{T}}\mathbf{y}$$

$$\mathbf{S} = \left(\mathbf{M}'^{\mathbf{T}}\mathbf{M}'\right)^{-1}\chi^2$$

$$\beta_i = \sqrt{S_{ii}}$$

with the solution of the linear problem $\hat{\mathbf{x}}$ containing the SCDs, $\mathbf{y}' = \text{diag}(\mathbf{w})y$ the weighted measurement spectrum, $\mathbf{M}' = \text{diag}(\mathbf{w})\mathbf{M}$ the weighted absorption structures to fit, $\beta_i$ being the estimated $1\sigma$ fit error of the results for each fitted parameter,
and $\chi^2$ the reduced chi-square.

## Appendix B: Evaluation of the water vapour absorption cross section

Figure A2 depicts intercomparisons between LP-DOAS and meteorological measurements of water vapour volume mixing
ratios (WVMR) at different altitudes (10 m, 40 m and 200 m) at the CESAR Tower for day- and nighttime during the Cabauw
Intercomparison of Nitrogen Dioxide Measuring Instruments 2 (CINDI-2) campaign. The results of the regression methods
indicate that for every altitude the LP-DOAS underestimates WVMR by around 17% during day and 11% during night. These
findings independently confirm the results of further LP-DOAS measurements taken at the Cape Verde Atmospheric Observa-
tory for which Lampel et al. (2015) observed an underestimation of around 8% when using the water vapour line lists from
HITRAN 2012. However when using the water vapour line lists from HITRAN 2008 Lampel et al. (2015) observe an excellent
agreement to the reference meteorological measurements at the observatory (see Table 8 in their paper). With the shortcomings
of HITRAN 2016 indicated by Wang et al. (2019) we conclude that it is most adequate to use the water vapour line list from
HITRAN 2008.





## Appendix C: Evaluation of methods for calculating water vapour scale height

The water vapour scale height can be calculated in different ways. Here we compare two different approaches: The first method is the calculation of the scale height via a weighted non-linear fit:

$$\min \sum_i \frac{(y_i - f(z_i, n_0, H_{nl}))^2}{\sigma_i^2}$$

$$f(z, n_0, H_{nl}) = n_0 e^{-\frac{z}{H_{nl}}}$$

where $y_i$ are the COSMIC profile data points, $f(z, n_0, H)$ is the approximation of the exponential function, and $\sigma_i$ is the inverse of the layer thickness at the observation $y_i$. The second method consists of summing up all the partial columns of the COSMIC

profile data until a defined threshold is reached, which in our case is 63% of the $H_2O$ VCD:

$$\frac{\int_0^{H_{sum}} n(z)dz}{\int_0^{\text{TOA}} n(z)dz} > 1 - \frac{1}{e} \approx 63\%$$

Figure A3 depicts the mean profile shapes calculated using both methods as well as the mean profile shape of the COSMIC data for different latitude bins for the year 2013 for which the sample size is largest. Further statistics of goodness are given in Fig. A4 (bias), A5 (mean absolute error) and A6 (standard deviation). In general the profile shapes of both methods agree

well with the COSMIC measurements, however Fig. A4 and Fig. A5 also reveal that the largest deviations occur in the lowermost troposphere, in particular for the southern polar regions. Nevertheless the profiles of standard deviations in Fig. A6 also demonstrate that both methods are able to well capture the vertical and temporal variations in the water vapour profile shape and that these variations are within the same range of the variation of the COSMIC profile data.

Figure A7 depicts histograms of the relative AMF deviation for both methods for selected latitude regions assuming nadir

viewing geometry and clear-sky conditions (like in Sect. 3.1 and Fig. 4). The peaks of the histograms for the sum method are close to the 0% line indicating very good agreement with AMF calculated from the COSMIC profiles. In contrast the histograms for the non-linear fit peak at values around 2% and show a broader distribution than the histogram of the sum method, thus revealing an inferior agreement to the reference AMFs. For cloudy-sky conditions (see Fig. A8), both methods are biased to smaller AMF-values (deviations of around -5%) for a cloud top height of 1 km, but for higher clouds both methods show

similar good agreement to the reference AMFs. Yet the variance in the AMFs for the sum method is much smaller than in the AMFs for the non-linear fit.

In summary the sum method is to be preferred because it provides more consistent results for clear-sky and cloudy-sky scenarios than the non-linear fit.



*Author contributions.* CB performed all calculations for this work and prepared the manuscript together with SB and TW and in collaboration
with all coauthors. SB developed the concept of the linearized retrieval scheme and CB and SB implemented most of the retrieval code. SD
helped with the McArtim calculations and HS helped with the tessellation of the TROPOMI $H_2O$ VCD orbit data to a regular grid. TW
supervised this study.


*Competing interests.* The authors declare that they have no conflict of interest.

*Acknowledgements.* We would like to thank ESA and the S-5P/TROPOMI level 1 and level 2 teams for their greak work on initiating and
realizing TROPOMI and for providing the respective data sets. We also thank NASA for providing MODIS and SSMI data and ECMWF for
providing reanalysis data. Furthermore we acknowledge UCAR and ROMSAF for providing SuomiNet and COSMIC data. We also would
like to thank Stefan Schmitt and Johannes Lampel from the Institute of Environmental Physics at the University of Heidelberg for performing
the analysis of the LP-DOAS measurements during CINDI-2 and for providing the WVMR results in a very useful format.




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



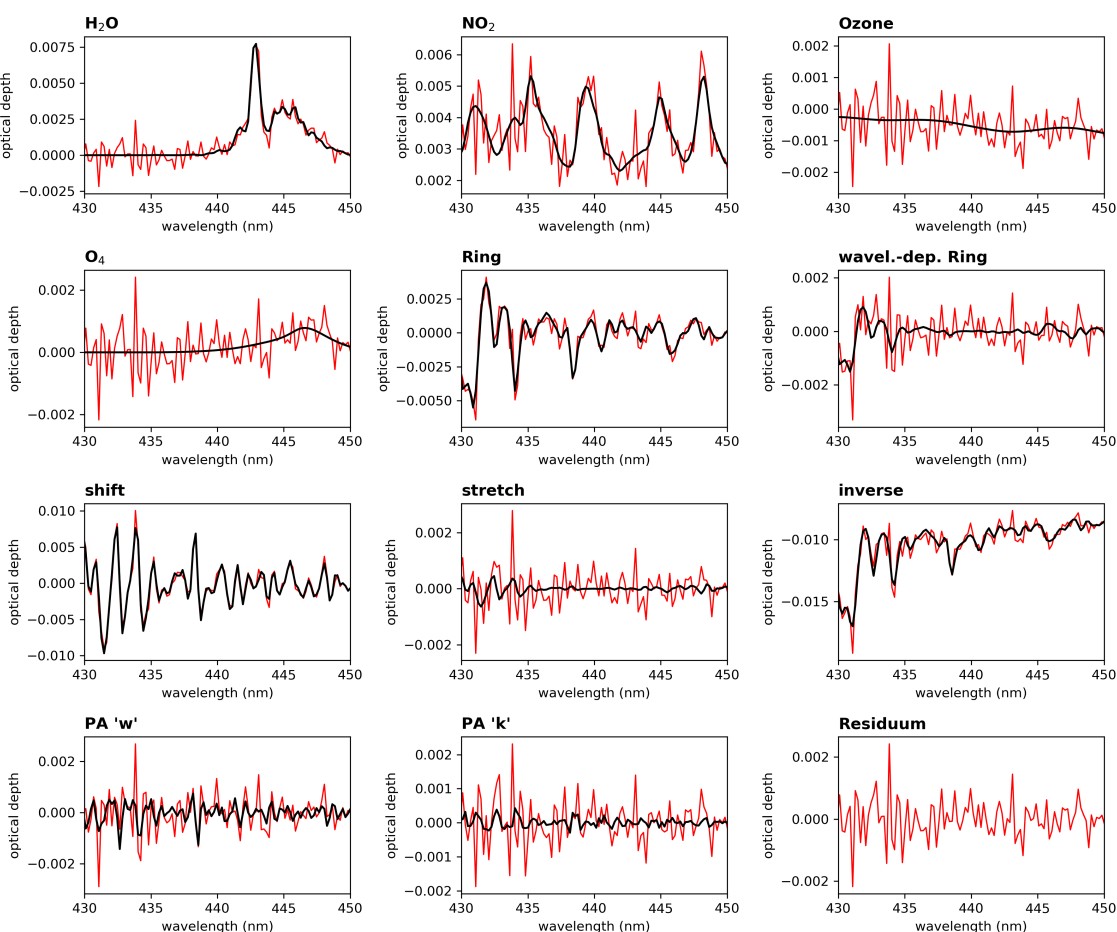

**Figure 1.** Example of a typical spectral analysis of a TROPOMI measurement spectrum (RMS: 0.6‰, orbit: 4088, 14.88°N, -2.59°E). The black line indicates the fit result for the respective trace gas and the red line indicates the residual spectrum and residual noise for each constituent.

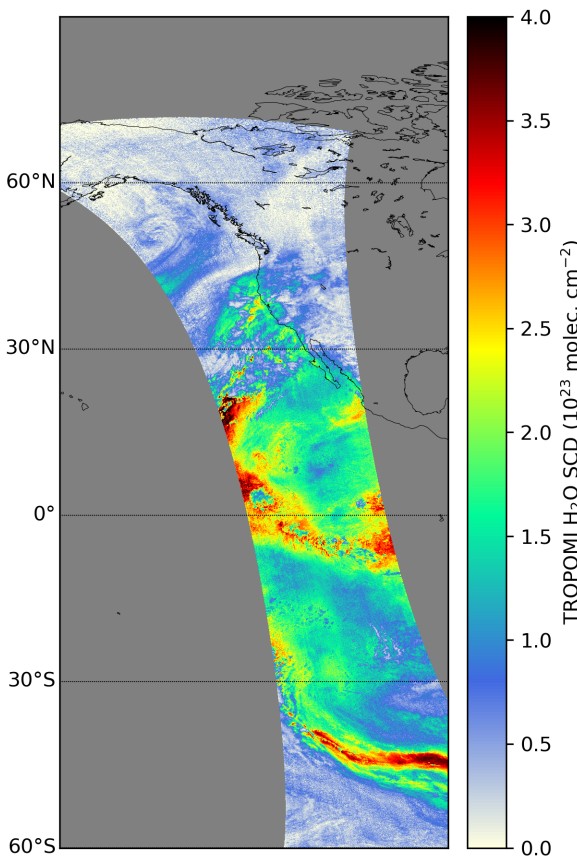

**Figure 2.** $H_2O$ SCD distribution retrieved from one TROPOMI measurements (orbit: 6930) on 13th February 2019 during an atmospheric river event at the Western US coast.





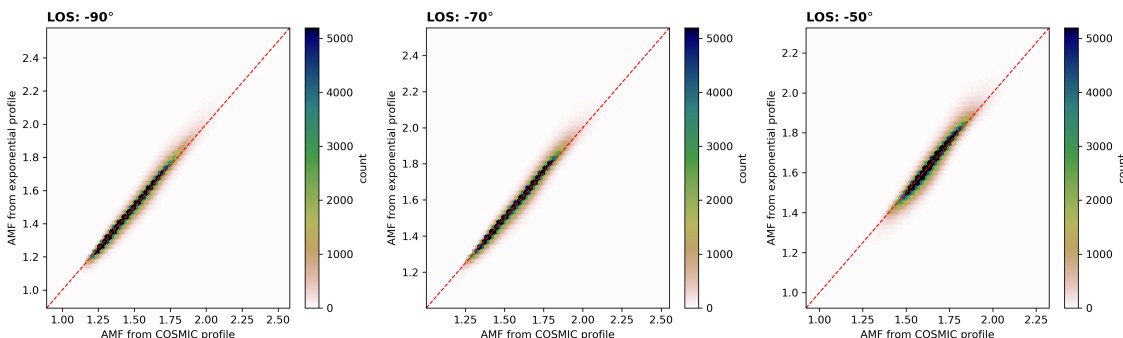

**Figure 3.** 2D histograms comparing synthetic AMFs (calculated via sum method) for different line of sights angles (−90°, −70° and −50°, from left to right) assuming clear-sky conditions. The color depicts the number of observations within one defined bin of the 2D histogram and the red dashed line represents the 1-to-1 diagonal.

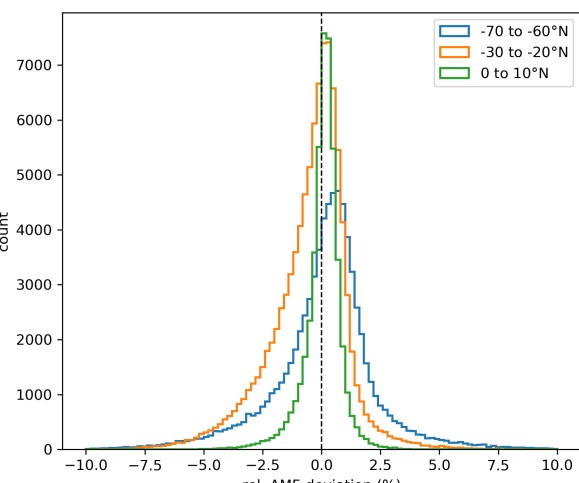

**Figure 4.** Histogram of the relative deviation of the calculated synthetic AMFs between exponential profile and COSMIC profile for selected latitude bins (0° to 10°N, -30° to -20°N, and -70° to -60°N) assuming clear sky conditions and nadir viewing geometry.



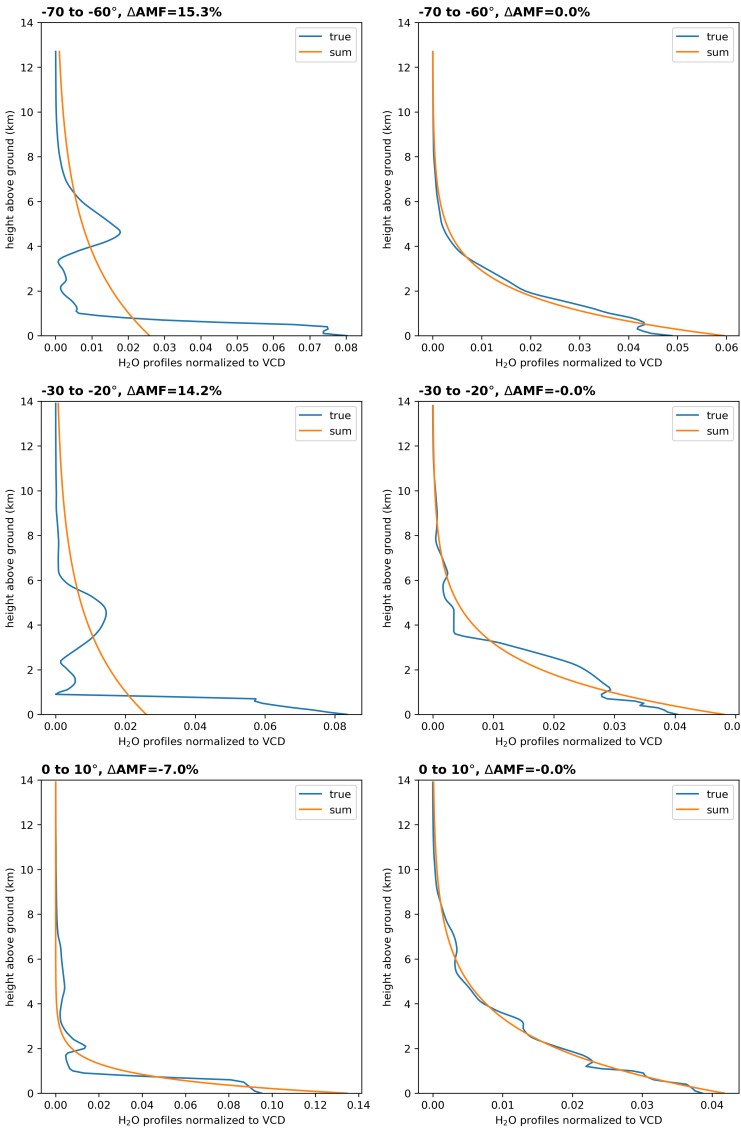

**Figure 5.** Comparison of profile shapes for selected latitude bins illustrating the maximal (left column) and minimal (right column) absolute relative AMF deviations. The blue line represents the "true" water vapour profile shape as measured from COSMIC and the orange line represents the exponential profile with a scale height $H$ calculated from the sum method.

**Figure 6.** Histograms of the relative AMF deviations between exponential profile and COSMIC profile for the same latitude bins as in Fig.4 for different cloud scenarios (cloud fraction 10, 20 and 50 % (left to right); cloud top height 1 km, 2 km and 5 km (top to bottom)) and a nadir viewing geometry.



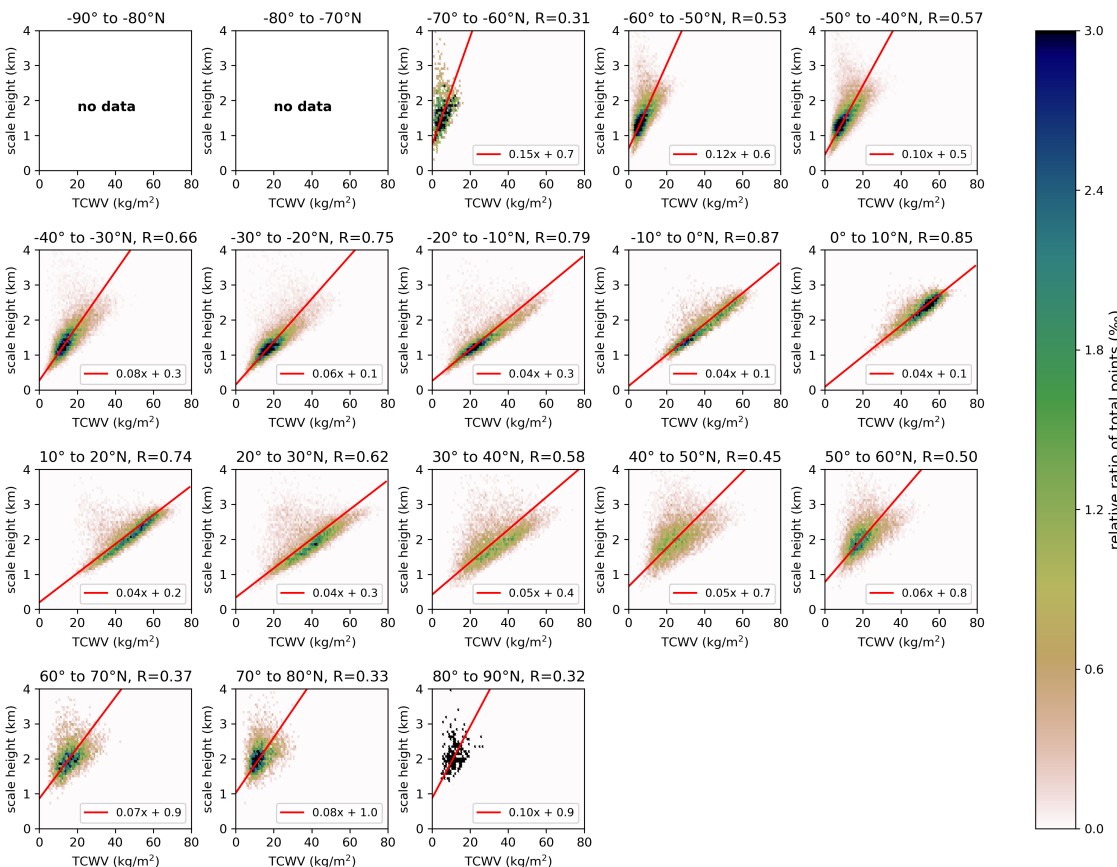

**Figure 7.** 2D histograms depicting the relation between calculated scale height and TCWV from COSMIC profiles for boreal summer (June, July, and August) only over ocean summarized in 10° latitude bins. Only latitude bins with a sample size of 1500 data points are illustrated. The color indicates the relative share of total points within one bin of the histogram and the red line indicates the fit results of the orthogonal distance regression with detailed results in the legend of each subplot. In addition the Pearson correlation coefficient for each data set is given in the title of each subplot.

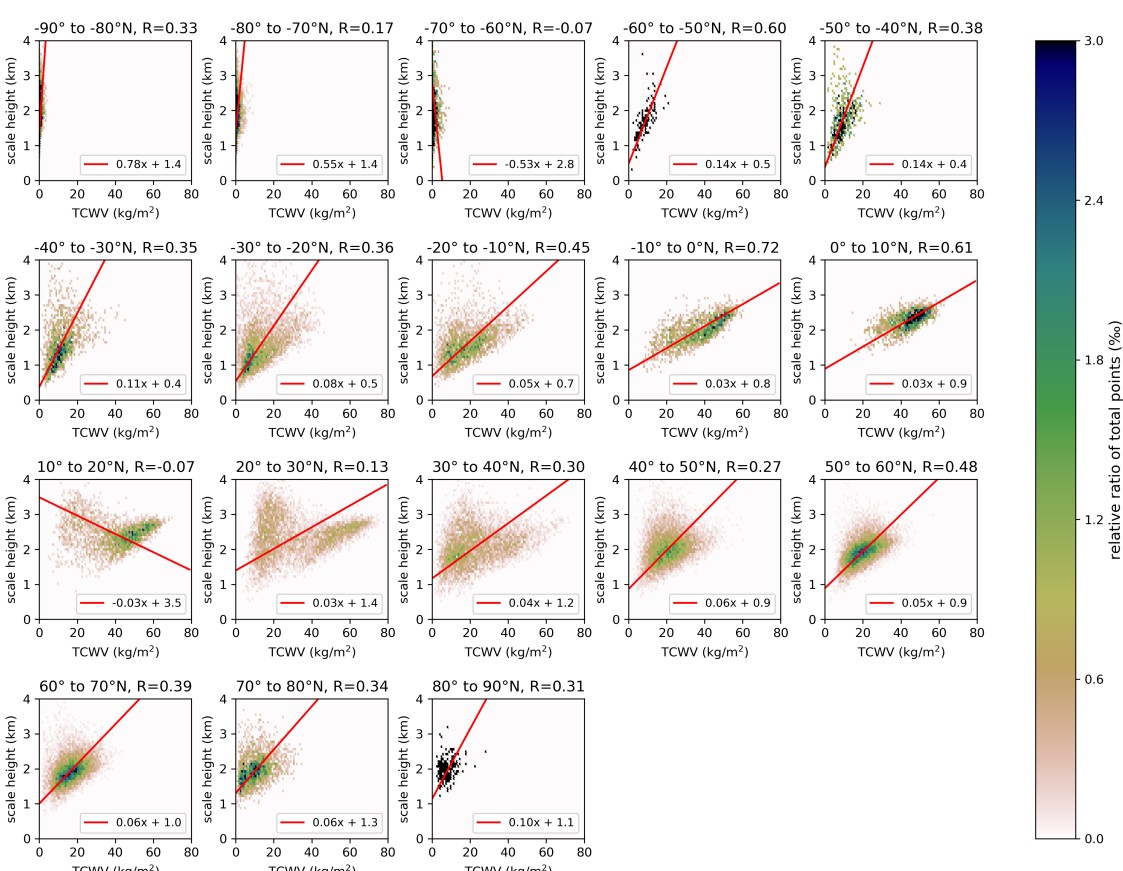

**Figure 8.** Same as Fig.7, but for data over land.



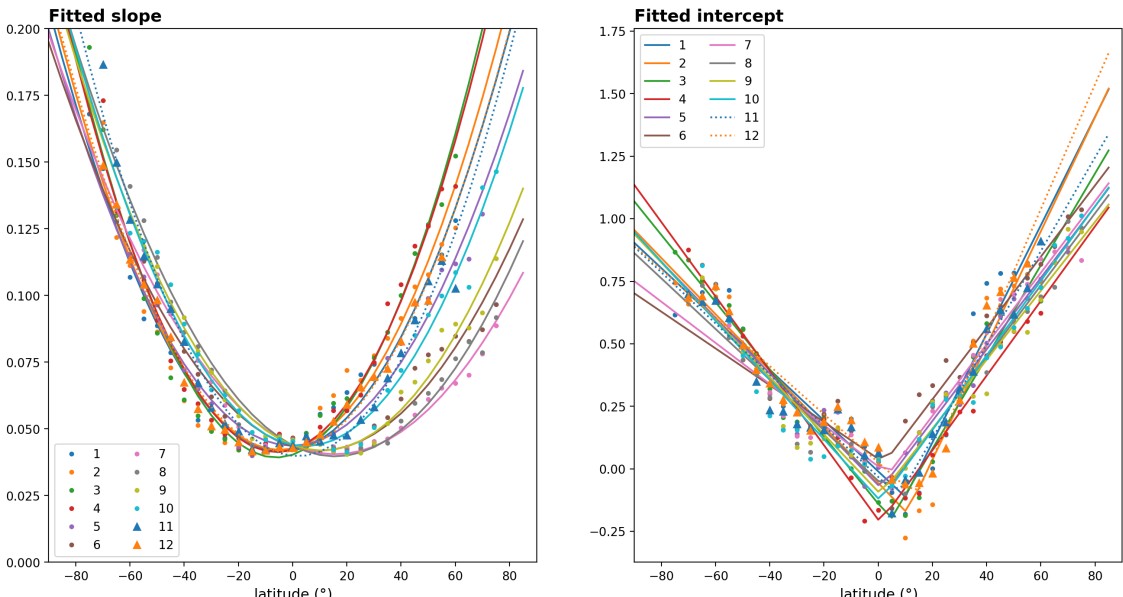

**Figure 9.** Summary of the results of the ODR fit between COSMIC scale height and COSMIC TCWV as a function of latitude and month for data over ocean. The left panel illustrates the fitted slopes, the right panel the corresponding fitted intercepts whereby the coloured points represent the fit results and the lines represent the approximations for $\alpha(\theta, t)$ and $\beta(\theta, t)$ for each month.



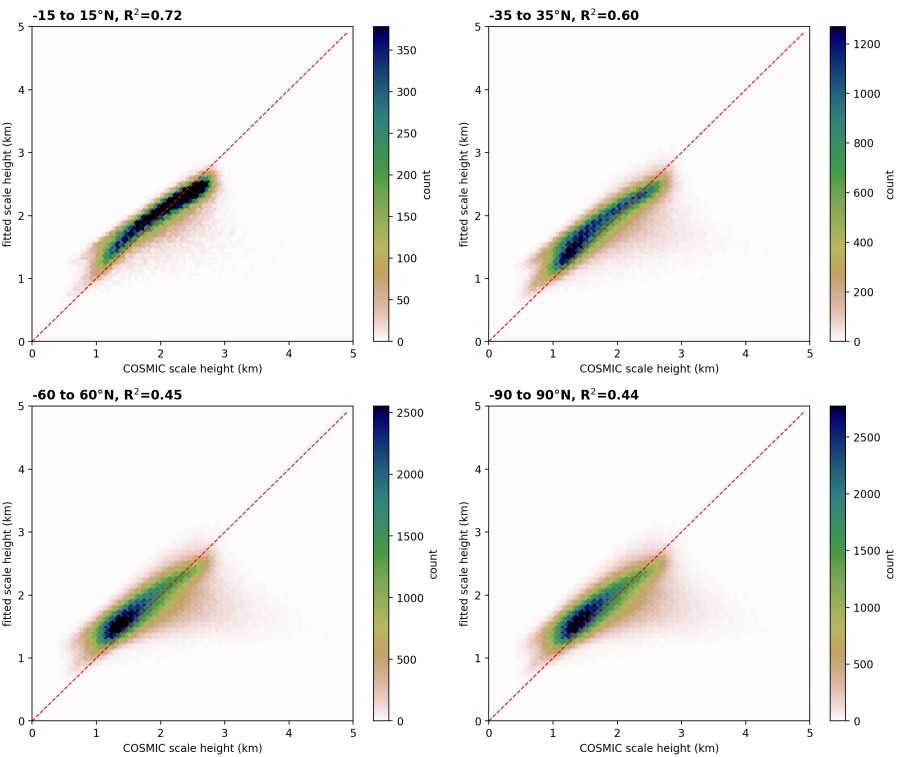

**Figure 10.** 2D histograms of the distribution between the parameterized scale height and the COSMIC scale height over ocean for selected global latitude zones.



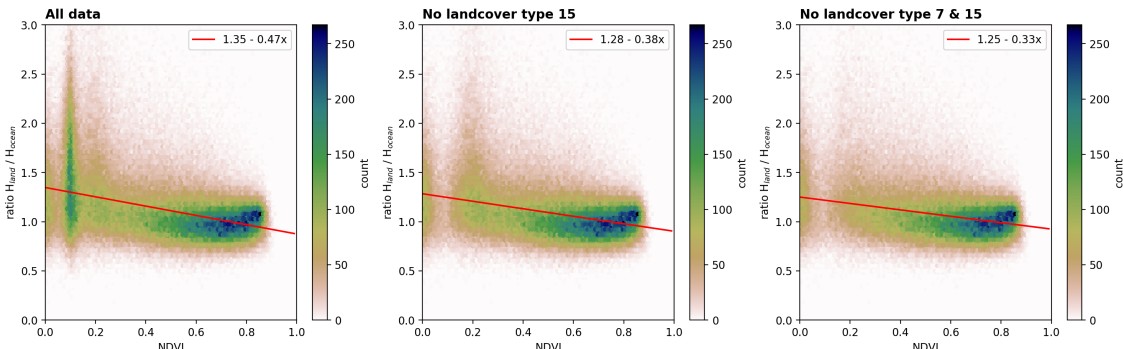

**Figure 11.** 2D histograms of the distribution between the ratio $H_{land}/H_{ocean}$ against the NDVI for different filtered data sets: The left panel includes all data points, the center panel includes all points except those with MODIS landcover type 15 (corresponding to deserts), and the right panel includes all points except landcover types 7 (corresponding to open shrublands) and 15. The red solid line represents the fit result using the Siegel algorithm with details of the fit results in the legends of each panel.

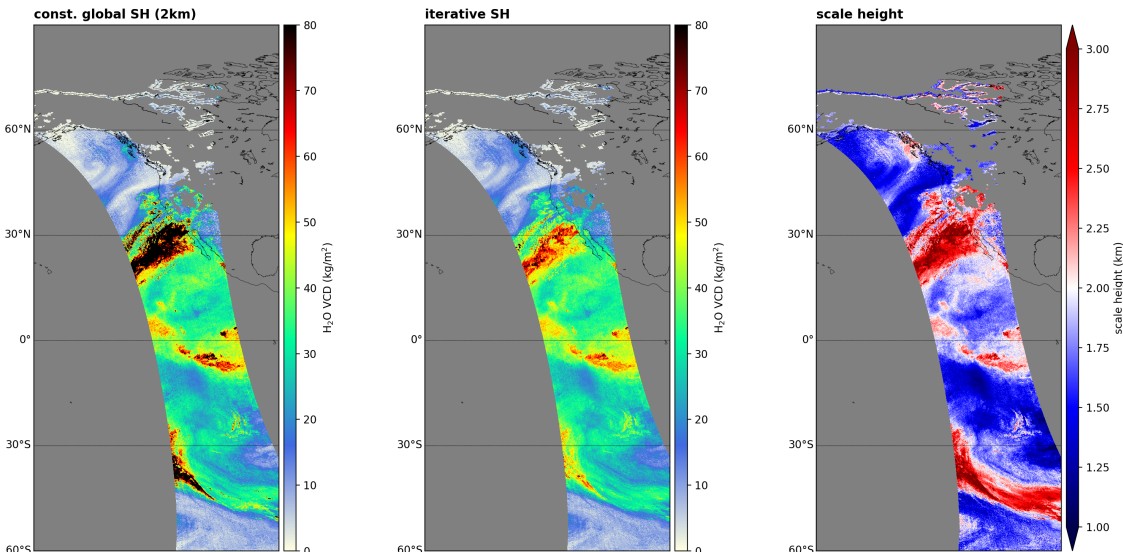

**Figure 12.** Comparison of the $H_2O$ VCD calculated using a global constant a priori profile shape of 2 km (left) and the iterative scale height method (center) for all-sky conditions. The right panel illustrates the water vapour scale height estimated within the retrieval's VCD conversion. All panels show an atmospheric river hitting the East Pacific/Western US Coast on the 13th February 2019. Invalid pixels are coloured grey.

**Figure 13.** Comparison of the H$_2$O VCD calculated using a global constant a priori profile shape of 2 km (left) and the iterative scale height method (center) for clear-sky (effective cloud fraction < 20%; top row) and cloudy-sky conditions (effective cloud fraction ≤ 100%; bottom row) with TCWV from SSMIS f16 (right) for the same scenary as in Figure 12. Invalid pixels are coloured grey. The solid black lines indicate the edges of the TROPOMI swath.





**Figure 14.** Comparison of the effect of different land albedo input data on the mean $H_2O$ VCD for boreal summer 2018 (top row: OMI monthly mean LER, middle row: OMI monthly minimum LER, bottom row: scaled MODIS Aqua blue surface reflectance). Only pixels with an effective cloud fraction smaller than 20% are included.



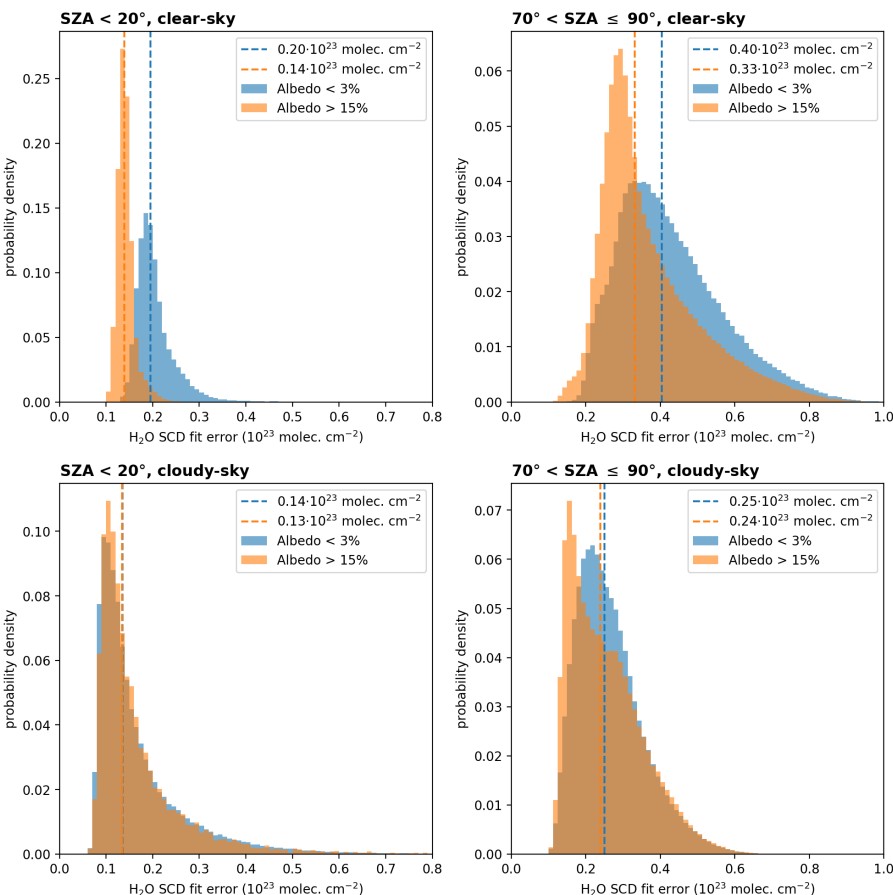

**Figure 15.** Histograms of the standard $H_2O$ SCD fit error distribution for small (SZA<20°, left panel) and large (70°< SZA≤90°, right panel) solar zenith angles for relatively small (<3%, orange) and high (>15%, blue) surface albedo values for clear-sky (cloud fraction < 5%, top row) and cloudy-sky (cloud fraction > 20%, bottom row) conditions. The coloured dashed lines represent the median of the respective distributions and their values are given in the legend of each panel.



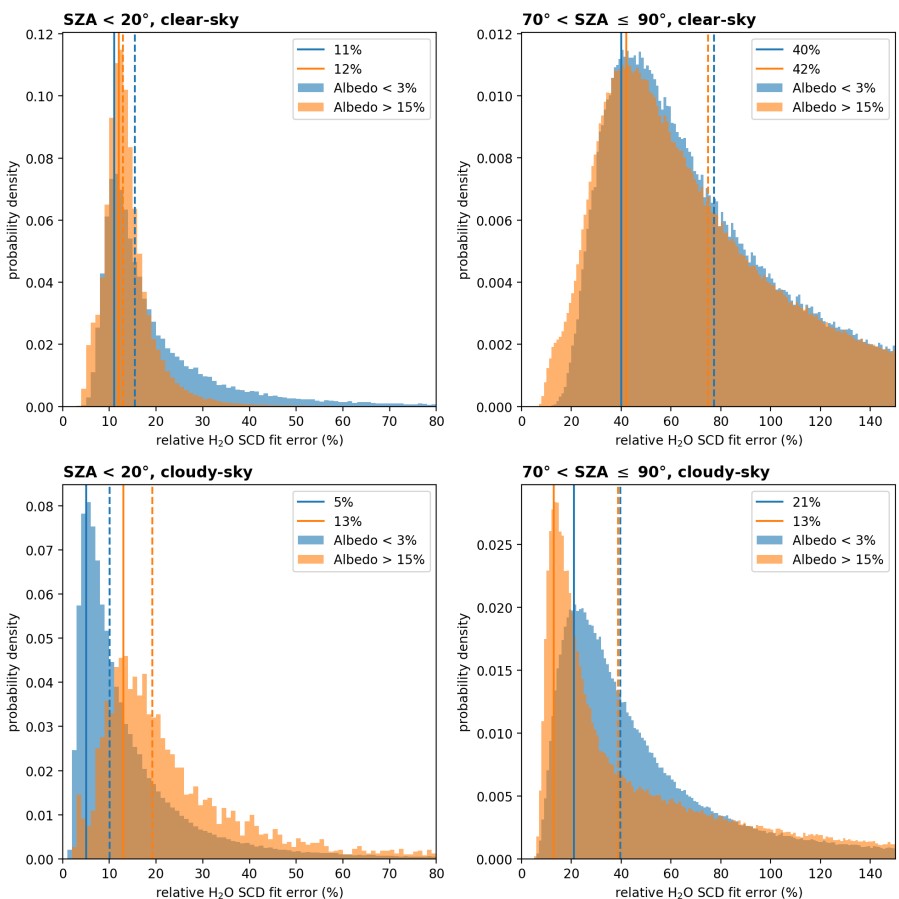

**Figure 16.** Histograms of the relative $H_2O$ SCD fit error distribution for small (SZA<20°, left panel) and large (70°< SZA≤90°, right panel) solar zenith angles for relatively small (<3%, orange) and high (>15%, blue) surface albedo values for clear-sky (cloud fraction < 5%, top row) and cloudy-sky (cloud fraction > 20%, bottom row) conditions. The coloured dashed lines represent the median of the respective distributions and the solid lines represent the location of maximal probability density (values given in the legend of each panel).



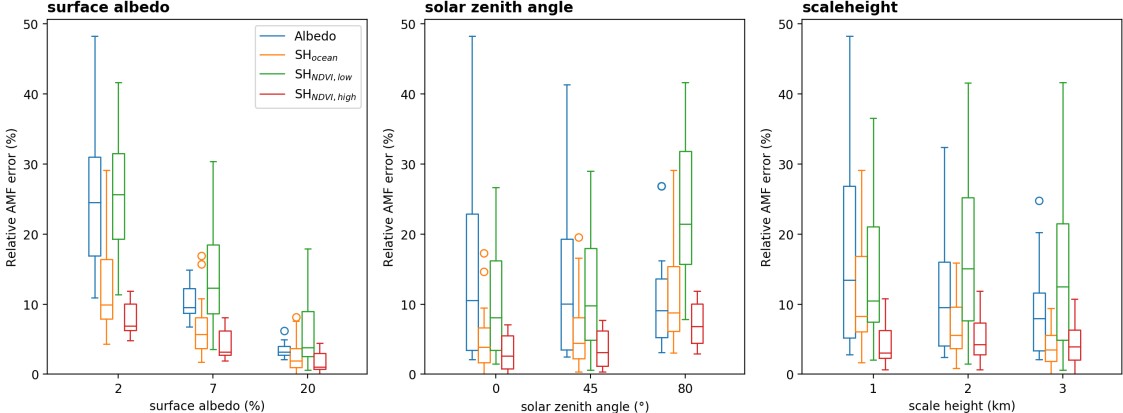

**Figure 17.** Box-whisker plots of the relative AMF errors for clear-sky conditions due to uncertainties within the retrieval's input parameters (blue: surface albedo, orange: scale height over high vegetation, green: scale height over low vegetation, red: scale height over ocean) according to the uncertainty assumptions in Tab. 6 and simulated for the standard scenarios of the surface albedo, solar zenith angle, and scale height given in Table 5.

.



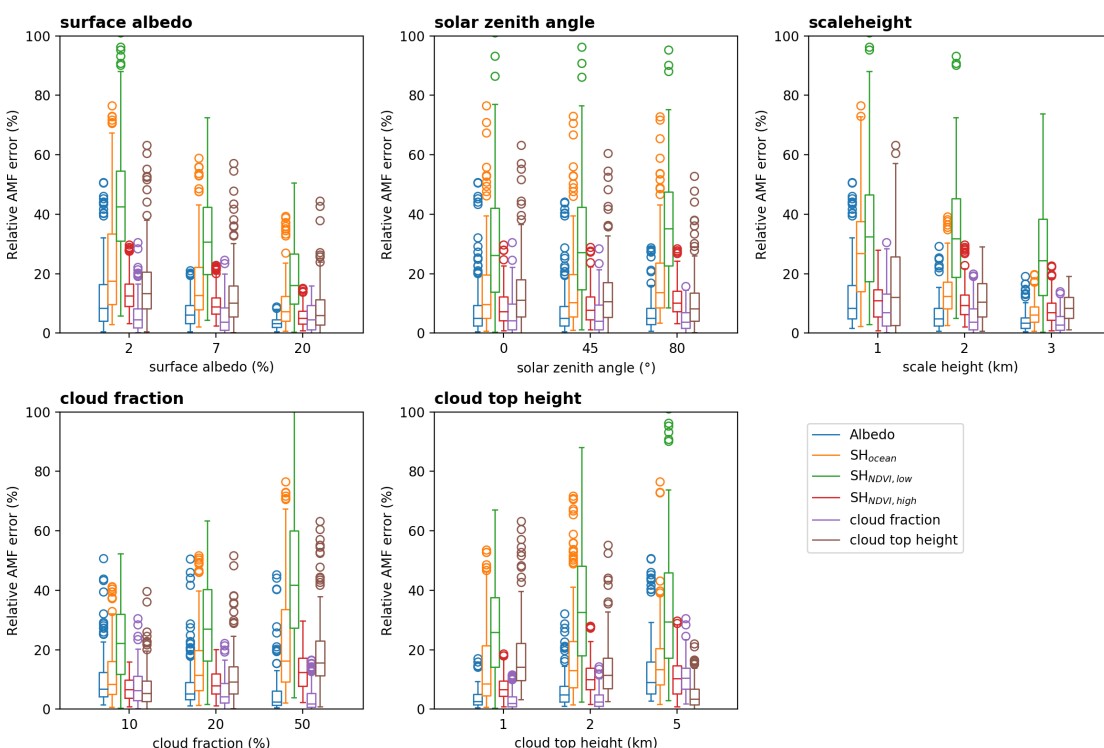

**Figure 18.** Box-whisker plots of the relative AMF errors for cloudy-sky conditions due to uncertainties within the retrieval's input parameters according to the uncertainty assumptions in Tab. 6 and simulated for the standard scenarios given in Table 5.

.

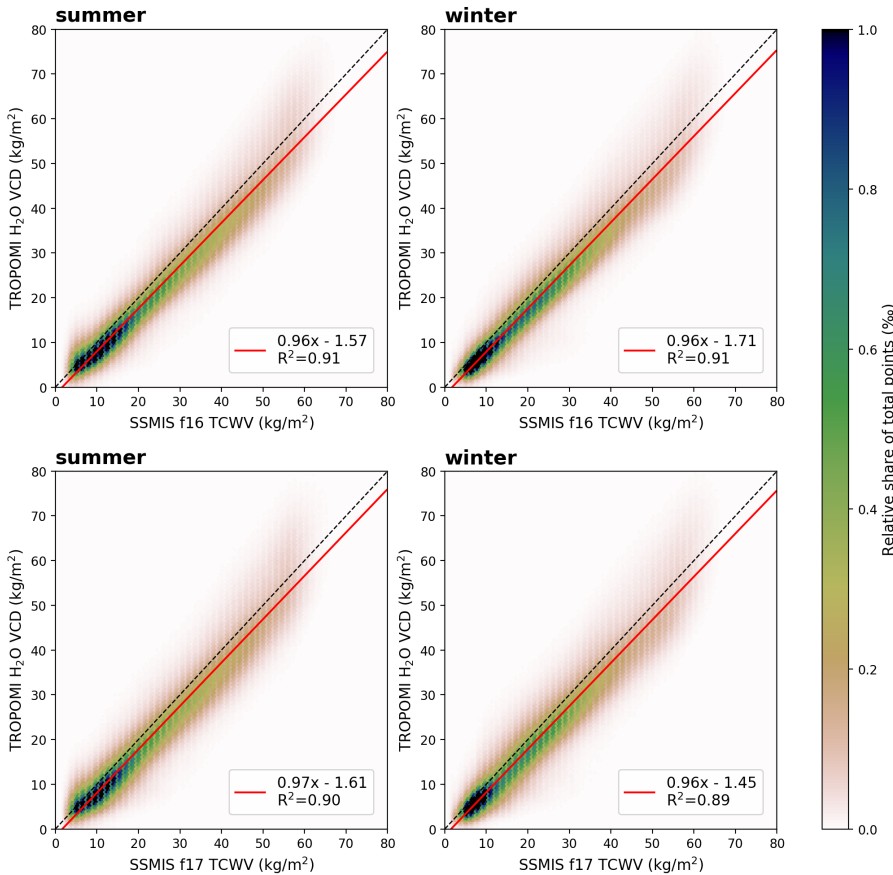

**Figure 19.** 2D histograms for the comparison between TROPOMI and SSMIS f16 (top row) and f17 (bottom row) for boreal summer (left column) and boreal winter (right column) where the color indicates the relative share of total data points. The black dotted line indicates the 1-to-1 diagonal and the red solid line represents the results of the linear regression. The parameters of the linear regression and the coefficient of determination are given in the box in each panel.

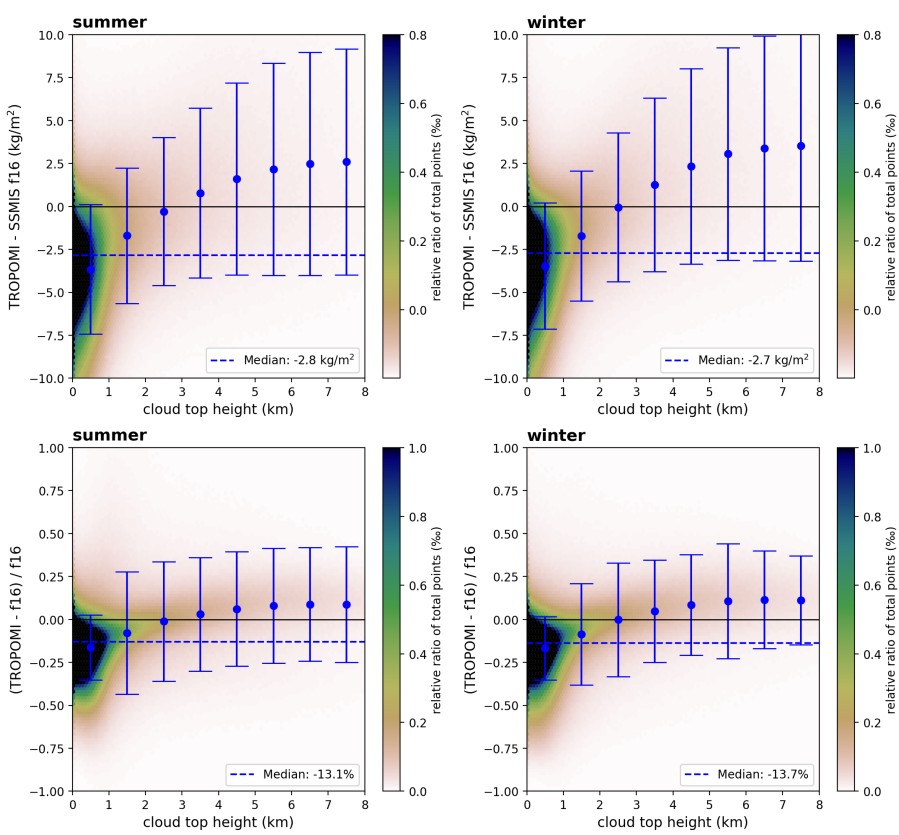

**Figure 20.** 2D histograms of the difference (TROPOMI-SSMIS f16, top row) and relative difference (TROPOMI-f16)/f16 (bottom row) as a function of the input cloud top height (CTH) for summer (left column) and winter (right column). The blue dashed line represents the median over the whole CTH range. The blue dots represent the median within a 1 km CTH and the error bars represent their respective $1\sigma$ standard deviation.



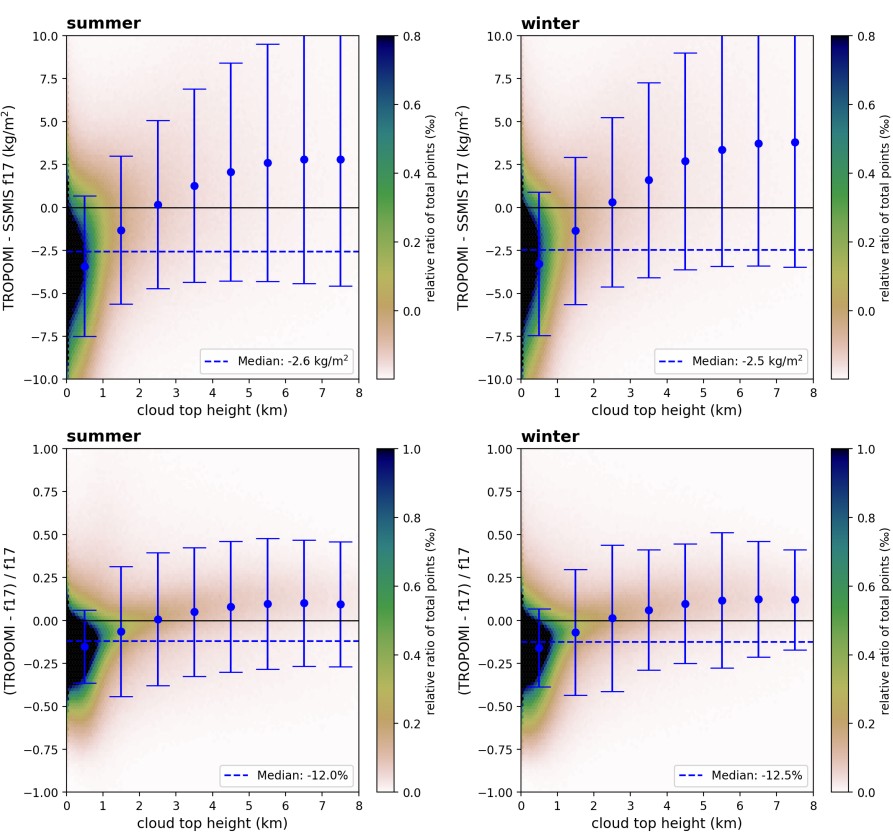

**Figure 21.** Same as Fig. 20, but for SSMIS f17.



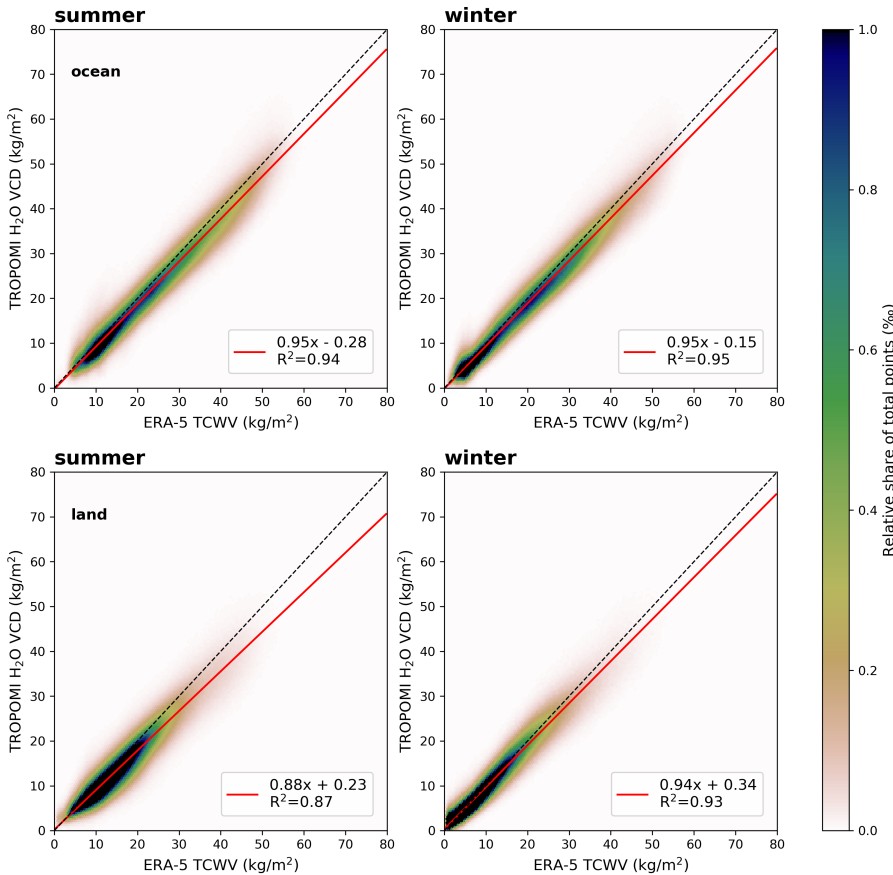

**Figure 22.** 2D histograms for the comparison between TROPOMI and ERA-5 for data over ocean (top row) and over land (bottom row) for boreal summer (left column) and boreal winter (right column) where the color indicates the relative share of total data points. The black dotted line indicates the 1-to-1 diagonal and the red solid line represents the results of the linear regression. The parameters of the linear regression and the coefficient of determination are given in the box in each panel.



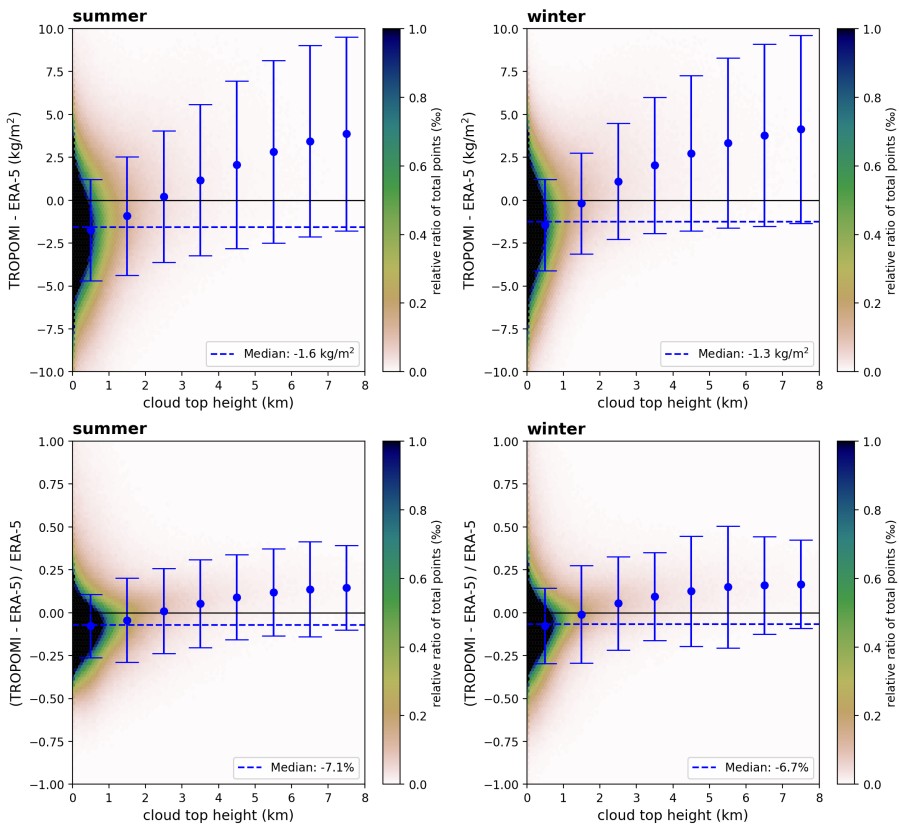

**Figure 23.** 2D histograms of the difference (TROPOMI-ERA-5, top row) and relative difference (TROPOMI-ERA-5)/ERA-5 (bottom row) as a function of the input cloud top height (CTH) for summer (left column) and winter (right column) for data over ocean. The blue dashed line represents the median over the whole CTH range. The blue dots represent the median within a 1 km CTH and the error bars represent their respective $1\sigma$ standard deviation.



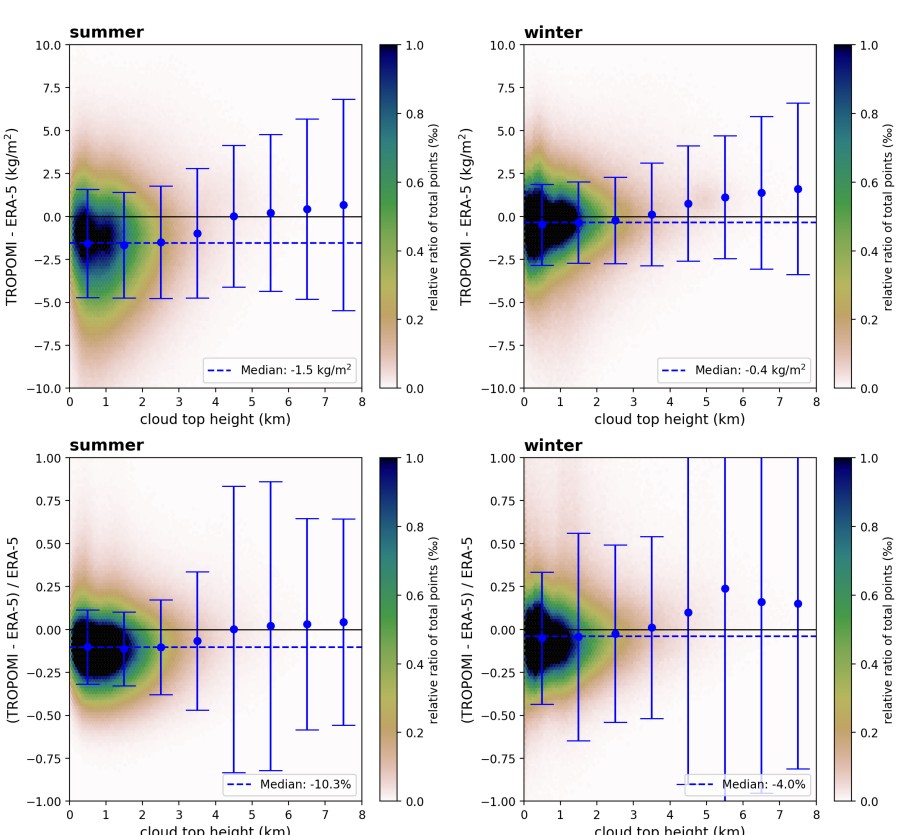

**Figure 24.** Same as Fig. 23, but for data over land.



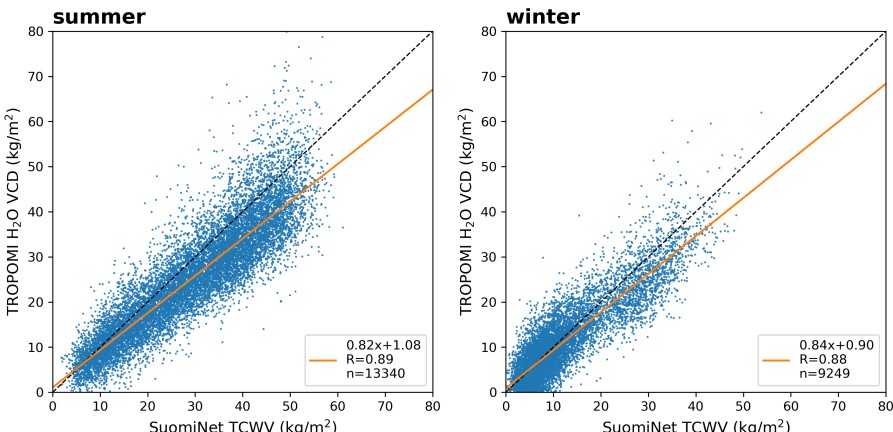

**Figure 25.** Scatter plots of the intercomparisons between TROPOMI and SuomiNet for boreal summer (left panel) and boreal winter (right panel). The black dashed line indicates the 1-to-1 diagonal and the orange solid line represents the results of the robust regression based on Siegel (1982). The parameters of the regression and the coefficient of correlation are given in the box in each panel.

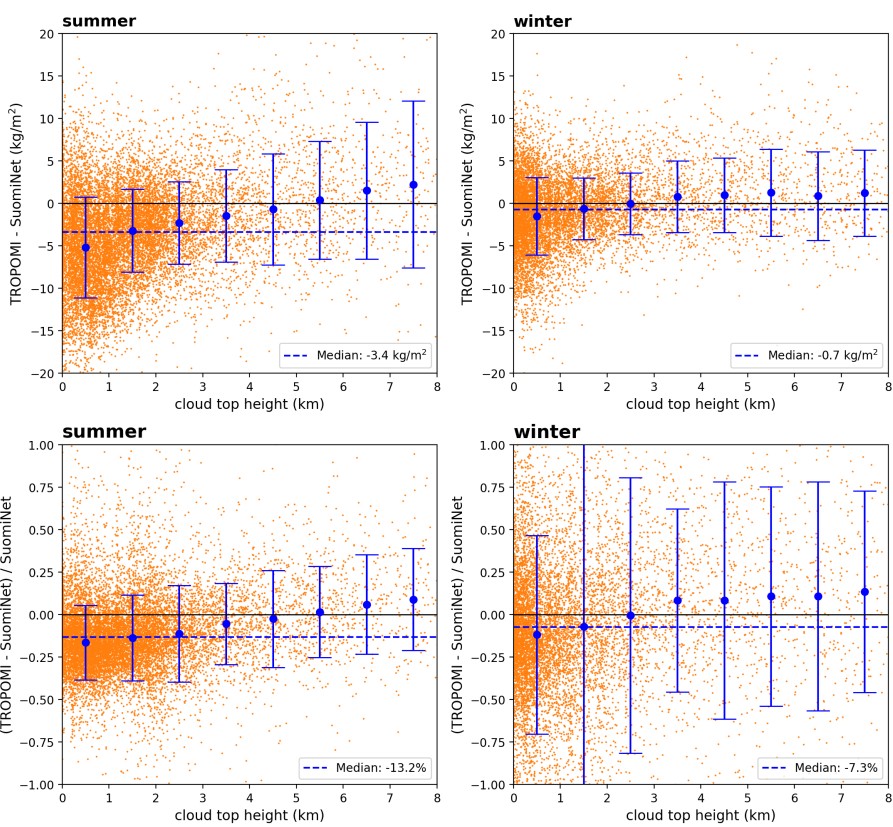

**Figure 26.** Scatter plot of the difference (TROPOMI-SuomiNet, top row) and relative difference (TROPOMI-SuomiNet)/SuomiNet (bottom row) as a function of the input cloud top height (CTH) for summer (left column) and winter (right column). The blue dashed line represents the median over the whole CTH range. The blue dots represent the median within a 1 km CTH and the error bars represent their respective $1\sigma$ standard deviation.




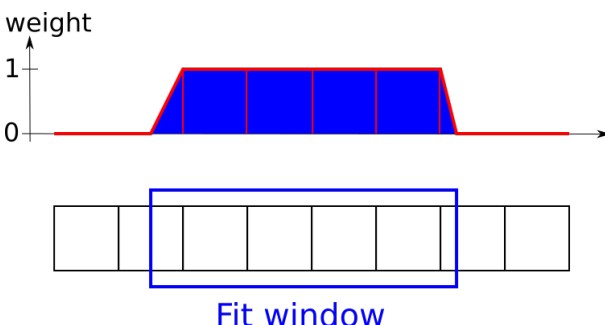

**Figure A1.** Schematic illustration of the weights used during the retrieval's spectral analysis.



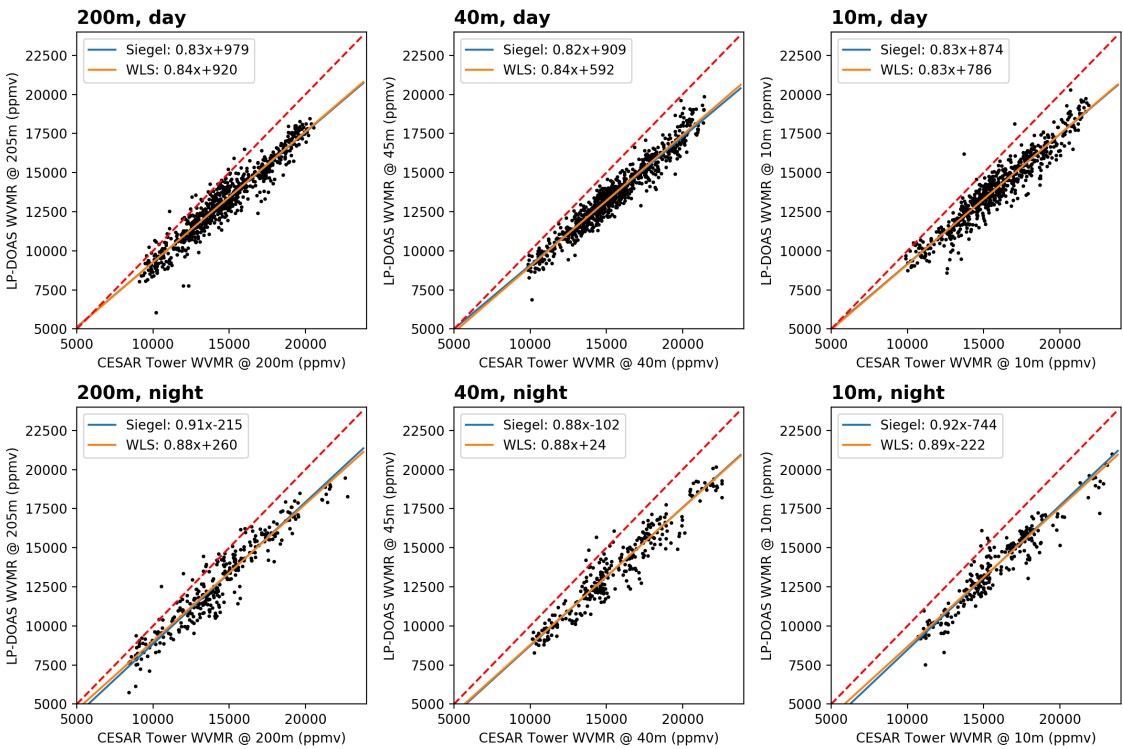

**Figure A2.** Scatter plots of water vapour volume mixing ratios (WVMR) derived from LP-DOAS measurements and meteorological measurements at different altitudes (10 m, 40 m and 200 m) at the CESAR Tower for day and night during the CINDI-2 campaign. Water vapour absorption cross sections have been calculated from HITRAN 2012 line list. The dashed red line represents the 1-to-1 diagonal, the solid blue line the results from the robust regression (Siegel, 1982), and the solid orange line the results from the weighted linear regression.





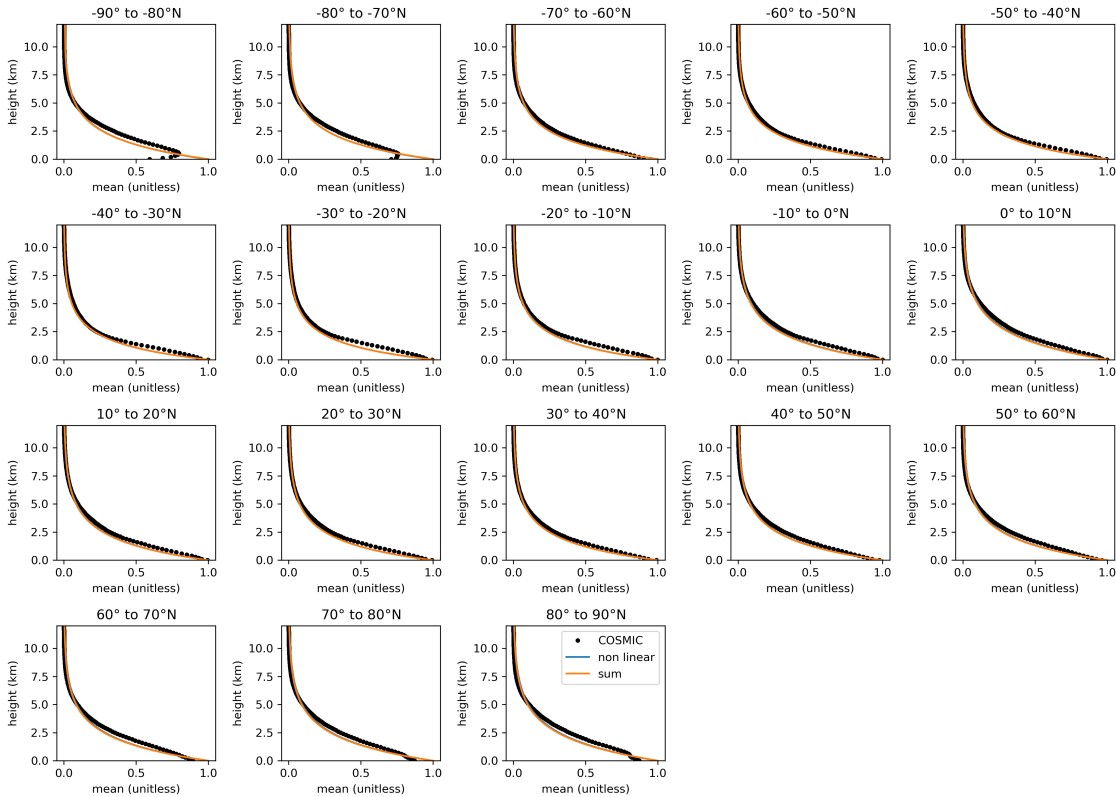

**Figure A3.** Mean water vapour profile shapes for data in 2013 for latitude bins of $10°$. The solid lines represent the results from the exponential scale height approaches (blue: non-linear fit; orange: sum method) and the black dots represent the COSMIC measurements.



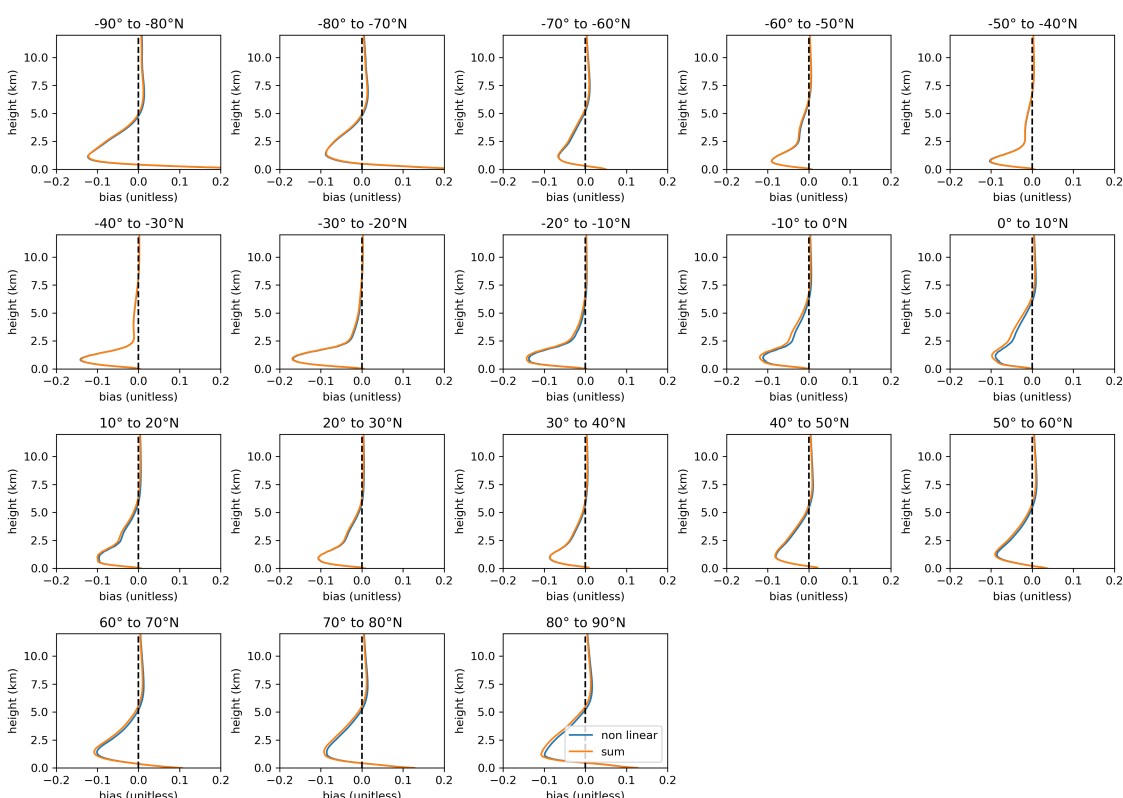

**Figure A4.** Bias of the profile shapes with respect to COSMIC profile shapes for the same data as in Fig. A3 (blue: non-linear fit; orange: sum method). The dashed black line represents the zero bias line.



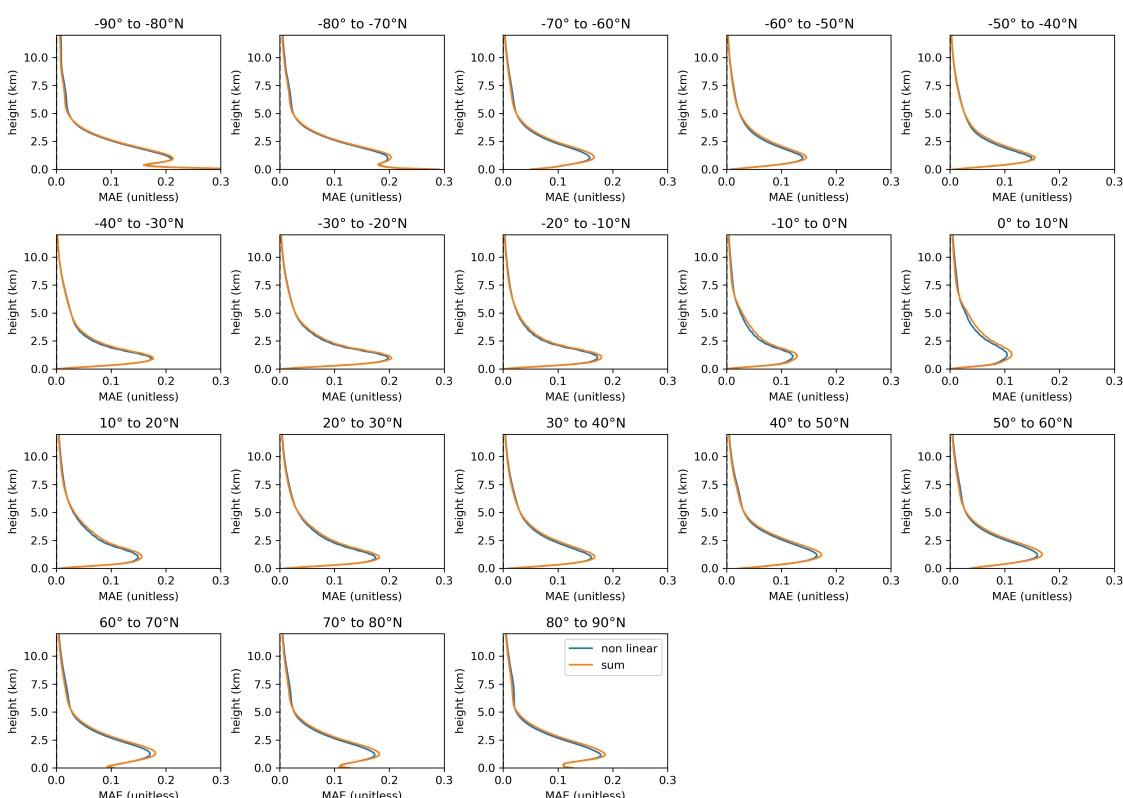

**Figure A5.** Mean absolute error of exponential profile shapes with respect to COSMIC profiles for the same data as in Fig. A3 (blue: non-linear fit; orange: sum method).



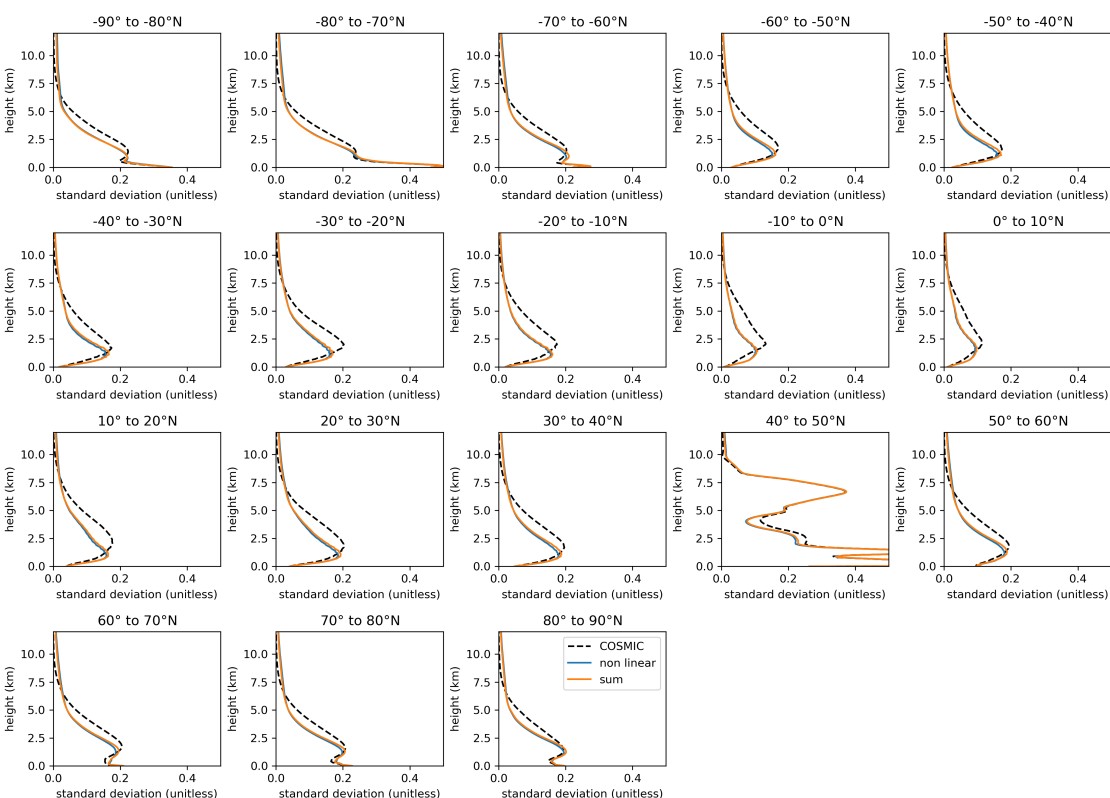

**Figure A6.** Standard deviations of the exponential and measured COSMIC profile shapes for the same data as in Figure A3. The blue solid line represents the results for the non-linear fit and the orange line for the sum method. The dashed black line represents the results for the COSMIC measurements.


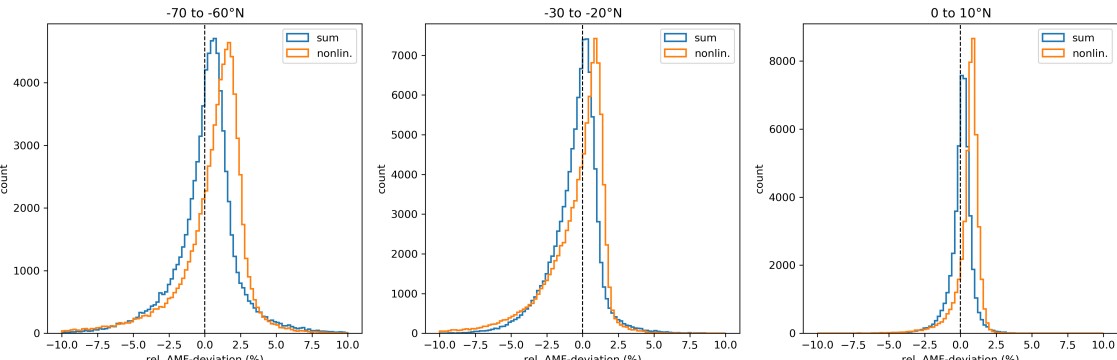

**Figure A7.** Histogram of the relative deviation of the calculated synthetic AMFs between sum method (blue)/non-linear fit (orange) and COSMIC profile for selected latitude bins (0° to 10°N, -30° to -20°N, and -70° to -60°N) assuming clear sky conditions and nadir viewing geometry.

**Figure A8.** Histogram of the relative deviation of the calculated synthetic AMFs between sum method (blue)/non-linear fit (orange) and COSMIC profile for latitude bin from -30°N to -20°N assuming different cloudy-sky conditions (cloud fraction 10, 20 and 50% (left to right); cloud top height 1 km, 2 km and 5 km (top to bottom)) and nadir viewing geometry.





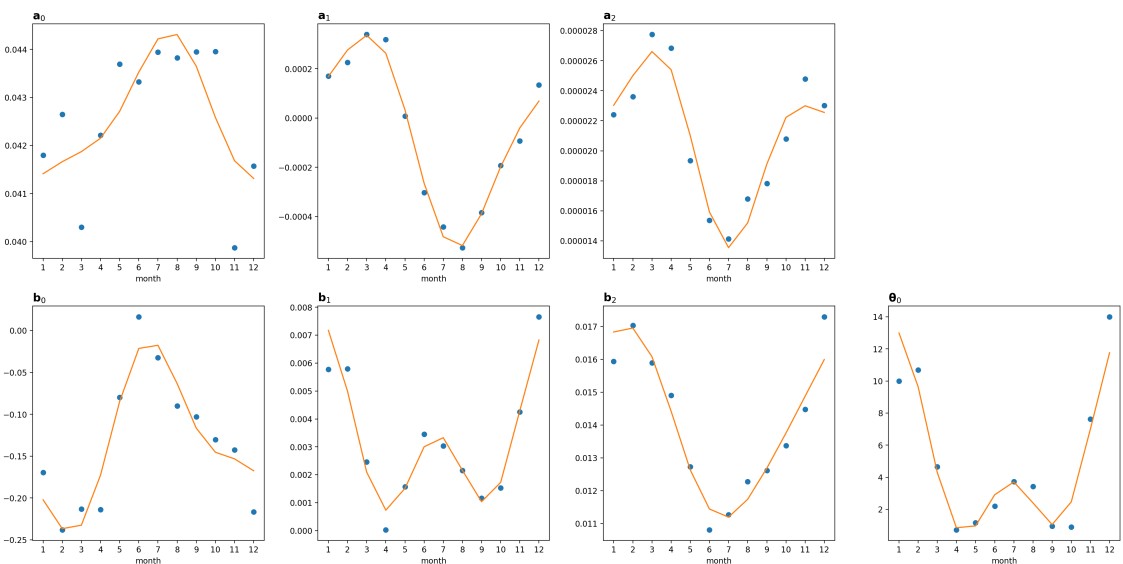

**Figure A9.** Monthly/seasonal dependence of the fit parameters $(a_0, a_1, a_2)$ and $(b_0, b_1, b_2, \theta_0)$ for the functions $\alpha(\theta, t)$ and $\beta(\theta, t)$ in Eq.(6).

**Figure A10.** 2D histograms for the comparison between TROPOMI and SSMIS f16 for July 2018 for different cloud fraction bins (left to right column) and cloud top height bins (top to bottom row). The color indicates the amount of points within one bin of each panel. The black dotted line indicates the 1-to-1 diagonal and the red solid line represents the results of the linear regression. The parameters of the linear regression and the coefficient of determination are given in the box in each panel.





**Figure A11.** Same as Fig. A10, but for SSMIS f17.





**Figure A12.** Same as Fig. A10, but for ERA-5 TCWV data over ocean.







**Figure A13.** Same as Fig. A10, but for ERA-5 TCWV data over land.





**Figure A14.** Scatterplots for the comparison between TROPOMI and SuomiNet for boreal summer 2018 for different cloud fraction bins (left to right column) and cloud top height bins (top to bottom row). The black dashed line indicates the 1-to-1 diagonal and the orange solid line represents the results of the robust regression. The parameters of the regression and the correlation coefficient are given in the box in each panel.





**Table 1.** DOAS fit settings for the H$_2$O slant column retrievals

| Parameter | Description |
| --- | --- |
| Fit window | 430 - 450 nm |
| Absorption cross sections | Water vapour, 296 K (Rothman et al., 2009) |
| | NO$_2$, 220 K (Vandaele et al., 1998) |
| | O$_3$, 243 K (Serdyuchenko et al., 2014) |
| | O$_4$, 293 K (Thalman and Volkamer, 2013) |
| Ring effect | 2 Ring spectra calculated from daily irradiance |
| Polynomial | 5th order |
| Pseudo-absorbers | intensity offset (inverse spectrum) |
| | shift & stretch (Beirle et al., 2013) |
| | ISRF parameter changes (Beirle et al., 2017) |



**Table 2.** Parameter list and nodes for the BAMF profile simulations

| Parameter | Nodes |
| --- | --- |
| Wavelength (nm) | 442 |
| Sensor altitude (km) | 720 |
| Surface altitude (km) | 0.0, 0.5, 1.0, 1.5, 2.0, 2.5, 3.0, 4.0, 5.0, 6.0, 8.0, 12.0 |
| Surface albedo | 0.0, 0.01, 0.02, 0.03, 0.04, 0.05, 0.06, 0.07, 0.08, 0.09, 0.10, 0.12, 0.14, 0.16, 0.18, 0.20, 0.25, 0.30, 0.40, 0.50, 0.80, 1.0 |
| Solar zenith angle (°) | 0, 10, 20, 30, 40, 50, 60, 65, 70, 80, 85, 87, 88 |
| Line of sight angle (°) | -90, -86, -82, -78, -74, -70, -66, -62, -58, -54, -50, -46, -42, -38 |
| Solar relative azimuth angle (°) | 0, 20, 40, 60, 80, 100, 120, 140, 160, 180 |





**Table 3.** Fit results of the robust regression between ratio of scale heights $H_{land}/H_{sum}$ and the NDVI for different filtered data sets.

| Data set | slope | intercept |
| --- | --- | --- |
| All data | −0.47 | 1.35 |
| No landcover type 15 | −0.38 | 1.28 |
| No landcover type 7 & 15 | −0.33 | 1.25 |





**Table 4.** Summary of the different error sources considered in the $H_2O$ SCD uncertainty.

| Source | Type | Parameter uncertainty | Estimated uncertainty in SCD |
|---|---|---|---|
| Absorption cross section | Systematic | 10% | 10% |
| DOAS fit error | Random | - | SZA<20°: $0.15 \times 10^{23}$ molec cm$^{-2}$ (~10%) |
| | - | - | SZA>70°: $0.30 \times 10^{23}$ molec cm$^{-2}$ (~30%) |


**Table 5.** Standard retrieval scenarios for the estimation of AMF error.

| Parameter | Values |
|---|---|
| Surface albedo | 2%, 7%, 20% |
| Solar zenith angle | 0°, 45°, 90° |
| Water vapour scale height | 1 km, 2 km, 3 km |
| Cloud fraction | 10%, 20%, 50% |
| Cloud top height | 1 km, 2 km, 5 km |
| Line of sight angle | -90° |


**Table 6.** Summary of different error sources considered in the AMF uncertainty.

| Parameter | Type | Parameter uncertainty | Source | Estimated uncertainty in AMF | |
| --- | --- | --- | --- | --- | --- |
| | | | | clear-sky | cloudy-sky |
| Surface albedo | Random+Systematic | 0.02 | Kleipool et al. (2008) | 5-25% | 5-10% |
| Scale height (ocean) | Random | 0.45 km | - | 2-10% | 5-20% |
| Scale height (NDVI, low) | Random | 0.73 km | - | 5-25% | 20-50% |
| Scale height (NDVI, high) | Random | 0.34 km | - | 2-7% | 5-15% |
| Cloud fraction | Random+Systematic | 0.05 | Veefkind et al. (2016) | - | 2-10% |
| Cloud top height | Random+Systematic | 0.5 km | - | - | 5-15% |