# Peer review of "Total Column Water Vapour Retrieval from S-5P/TROPOMI in the Visible Blue Spectral Range"

_Atmospheric Measurement Techniques, 2019_

## Referee Comment (RC1) · Anonymous Referee #1 · 10 Feb 2020

Review of Total Column Water Vapour Retrieval from S-5P/TROPOMI in the Visible Blue Spectral Range by Christian Borger et al.

This paper presents a study where the authors have developed an algorithm to retrieve total column water vapour (TCWV) from 'blue' portion of the visible spectrum band on Sentinel 5-P/TROPOMI, and make comparisons with some validation sources.

This is an interesting study, and is a good example of exploiting the potential satellite data beyond that which was intended by the developers of TROPOMI. It is certainly of interest to the AMT community, and I recommend publication, if the following points and concerns are addressed.

General Comments

[Figure]

In general, the paper lacks context to the wider TROPOMI picture, and only very briefly states the justification for why the authors attempt TCWV retrieval in the blue part of the visible spectrum. Even here, this discussion is in relation to the 'red' portion of the visible spectrum (where the authors have experience), and not products from other spectral regions (e.g. the SWIR). Especially since the errors can get quite high with these retrievals (up to 50%), I think the authors need to discuss the added benefit of retrievals from the 'blue' band to TROPOMI. The authors very briefly give some discussion to this in the conclusions (processing time, etc), this needs to be expanded upon.

The authors identify TCWV products from other spectral regions on other instruments (e.g. TIR from AIRS), but there is no discussion on how TCWV from the 'blue' band compare to the TCWV products from these other instruments and spectral regions. This discussion needs to be included into the paper.

Section 1 – The Introduction.

Context is lacking in this section, it is unclear as to the advantages of retrieving TCWV from this particular spectral region. Given the other spectral bands and satellites that the authors mention, and not just the 'red portion' of the spectrum.

Indeed, the authors have not referenced the study covering water vapour retrieval from the 2305-2385 nm TROPOMI spectral band (Schneider et al., 2020), which would have been in the AMT discussion forum at time of submission. They have also missed water vapour retrieval studies from the Shortwave Infrared (SWIR) in general, this is very strange since the SWIR gives good sensitivity to the surface e.g. (Trent et al., 2018), and surface sensitivity is clearly one of the key selling points for the 'blue' region.

Section 3 – A priori water vapour

While the aim of this section is clear, the story of how the methods that are used are a little unclear, and the general message gets a little lost in the details. Here in

section 3, I think a few sentences that summarises the processes that are undertaken in sections 3.1 and 3.2 would be useful. Readers may well be unfamiliar with COSMIC and ROMSAF, and some background would be beneficial here. Given the importance of the a priori profile on the results, some discussion on the biases associated with using COSMIC as a base would be welcome.

Section 6 – Validation study.

More detail about the SSMIS and SuomiNet data are needed here, fundamentally why did the authors choose these measurements for their validation, and why are they appropriate for TROPOMI inter-comparisons?

In the introduction the authors identify some specific advantages of retrieving TCWV from the blue part of the spectrum as opposed to the red, which has been done previously. To me, the validation section would have been a good opportunity to compare against TCWV data from the red part of the spectrum, not necessarily from TROPOMI, but from other satellites. Indeed comparisons from other 'blue' TCWV measurements from OMI as identified by the authors would have been useful, to identify any potential differences between the spectral regions. Or, please justify their exclusion from the study. Validation is largely done using TCWV retrieved from microwave instruments, are there any particular biases associated with TCWV retrievals, when compared against the visible band?

Other

With regards to the English, the paper could be improved given a review by a native English speaker. I will not go into detailed specifics in this review, but this paper would benefit from a greater use of punctuation (i.e. commas).

There are a large number of figures in this paper, and while I applaud the efforts of the authors for the detail they have gone into, 26 figures + 14 in the appendices is unneeded for a journal paper, which should only be showing the key highlights. I

suggest to the authors to place some of these figures in supplementary materials.

In addition, on some of the Figures the axis labels and legends are quite small (e.g. Figure 5), I recommend that the authors increase the font size.

Additional thoughts

I note that the authors have stated that their dataset is available on request, I would strongly encourage them to place their dataset into an online repository.

Specific comments.

On p3, line 75. The authors state that the absorption is weak in the fit window, and hence the line lists vary. I am not convinced by this argument, to me just because absorption is weak the uncertainty of the spectral lines shouldn't necessarily be higher, does HITRAN state this? What exactly is different between the databases, is it the position of the lines, the number of lines?

P4, line 91. Why are the data shown from Figures 1 and 2 based on different orbits? It'd be more consistent to show the same orbit results.

P5, Equation (4) – The calculation of atmospheric refractivity is highly dependent of air pressure and air temperature. Therefore the scale height for the water vapour profiles is highly dependent on knowledge of these factors. How accurate is this knowledge, and how sensitive is the calculation of scale height to errors in the knowledge of air pressure and air temperature?

P5, line 143. Why 63% (assuming this is total water vapour up to 150 hPa)? Is this because there is not much water vapour above this point? If so, this should be stated.

P5, line 147. Why 7%?

P6, I think Figures 5 and 6 could be moved to the appendices or supplementary material. For Figure 5, it is very unclear to me exactly what causes the 'bad' profiles, as opposed to the good profiles, this needs to be expanded upon.

P7, line 187. How are ocean and land differentiated? What about heterogeneous scenes or lakes?

Figure 14. The authors claim that there is a distinct separation of the H2O VCD between land and ocean, which vary between albedo versions. I did see it eventually, but it is quite subtle, this might be represented better if focused upon? Also, I don't feel that the inclusion of this Figure is particular beneficial to the paper, and can be placed in supplementary material.

P10, section 5. There is no discussion on instrumentation errors such as ILSF biases and radiometric errors. Can you comment on the impact of these?

P11, line 316. Is the spectroscopic uncertainty not as significant as any of these other factors? It would be useful to state the impact of this uncertainty in relation to the others. These are mentioned in the summary, but would be useful in the main text as well.

P15, summary and conclusion. Can you comment on the uncertainty of the retrievals (you say up to 50%) in comparison to TCWV retrievals from other spectral bands.

P15, lines 469-471. Here the authors state that the retrieval allows for a fast execution of large datasets. This to me is one of the key benefits of the retrievals in this waveband, however this is the first time that it has been mentioned in the manuscript. A brief discussion on processing times, and comparisons with other TCWV products would be beneficial.

Technical P2, line 45 – A reference should be added for the increase in TROPOMI spatial resolution to 3.5x5.6.

P3, line 83 – Band 4 is mentioned for the first time, please identify what TROPOMI Band 4 is.

A number of the equations appear to be missing equation numbers, e.g. the AMD calculations p4, lines 106, 111.

P5, line 120 – Could you provide an explicit statement for what is meant by scale height please.

P5, line 130 – Please define ROMSAF.

P5, line 135 – GPS RO allows to retrieve profile information -> GPS RO allows for the retrieval of profile information

P15, line 468 – and Sentinel 5

P19, line 534 – greak -> great

References

Schneider, A. et al. (2020) 'First data set of $H_2O$/HDO columns from the Tropospheric Monitoring Instrument (TROPOMI)', Atmospheric Measurement Techniques, 13(1), pp. 85–100. doi: 10.5194/amt-13-85-2020.

Trent, T. et al. (2018) 'Observing Water Vapour in the Planetary Boundary Layer from the Short-Wave Infrared', Remote Sensing, 10(9), p. 1469. doi: 10.3390/rs10091469.

---

## Short Comment (SC1) · 12 Feb 2020

**Eamon Conway**

eamon.conway@cfa.harvard.edu

Received and published: 12 February 2020

We have a couple of comments on the way different editions of the HITRAN database are represented in this paper.

1. Regarding statement in Line 75. "The molecular absorption by water vapour within our fit window is relatively weak and hence the modelled line lists vary strongly from HITRAN 2008 to HITRAN 2012 (Rothman et al., 2013) and to HITRAN 2016 (Gordon et al., 2017)."

Please see attached figure, which shows the absolute absorption cross-sections of HITRAN2008 and HITRAN2012 plotted on a linear scale, all isotopologues included. The cross-sections were all calculated using the HITRAN API with a temperature of

288K, 1 atm, 0.001 cm-1 resolution and a 50 cm-1 wing. The data sets appear similar and do not seem to "vary strongly". The residuals at 446.2 nm, 444.2 nm and 444.5 nm are due to improvements in calculated line positions in HITRAN2012. The line centers are slightly shifted, but they are present in each set. The HITRAN2012 line list is also more complete than HITRAN2008, note the extra HITRAN2012 weak absorption around 440 nm. Can the authors provide any evidence that suggest the data sets "vary strongly" in 430-450 nm?

2. Regarding statement in Line 77. "Lampel et al. (2015) found out that HITRAN2012 underestimates the water vapour concentration derived from Long Path DOAS observations by approximately 10% and that the previous version HITRAN2008 agrees better to the reference measurements."

Within the discussion section in Lampel et al. (2015b), they found residuals in all their windows to be reduced by going from HITRAN2009 to HITEMP2010/HITRAN2012 cross-sections. (HITRAN2008 is HITRAN2009: the article was online in 2009 but the edition of the database is HITRAN2008.) Lampel et al. (2015b) also say "development of water vapour absorption compilations from HITRAN 2009 to HITEMP/HITRAN 2012 results in a better fit of the measurement data". This is not in line with what is claimed here? Can the authors please verify?

3. Appendix B. The authors state that Lampel et al. (2015b) found the HITRAN2012 cross-sections to underestimate water vapor mixing ratios by 8% in 430-450 nm, while for HITRAN2008, the results are in excellent agreement with the meteorological station. The meteorological station is quoted by Lampel et al. (2015b) to have a 5% uncertainty on their value of humidity and a 2% error on the temperature. Lampel et al. (2015b) then proceeds to state "absolute differences of the cross-sections shown in Table 3 cannot be absolutely validated with sufficient precision". It therefore may not be true that HITRAN2008 is 'superior' to HITRAN2012 and 8% deviation is very close to the uncertainty of the measurement. Does this provide sufficient evidence to support the use of HITRAN2008 over HITRAN2012?
**C4**

Fig. 1. Absorption cross sections for the HITRAN2008 and HITRAN2012 water vapor line lists

computed at 288K, 1 atm, 0.001 cm\$^{-1}\$ resolution and 50 cm\$^{-1}\$ wing.

---

## Referee Comment (RC2) · Ruediger Lang (Referee) · 24 Feb 2020

The paper by Borger et al., on total column water vapour retrievals from Sentinel-5P (S5P) is demonstrating the large potential of water vapour retrievals in the blue and visible spectral range to yield an accurate estimate of the total water vapour column (TWVC) largely independent from model data and capable to cover all surfaces. This type of TWVC product from instrumentation like GOME-1/2, SCIAMACHY, OMI and TropOMI, therefore serves as an important product for the evaluation of (re-)analysis NWP model data output.

The paper by Borger et al. is overall well written and structured and apart from presenting the very first results of this type of TWVC product from the S5p mission the

paper also presents an interesting and novel approach of employing sub-column water vapour profile information to TWVC retrievals, via a parameterization of the water vapour atmospheric scale-height for various conditions, like surface type and different observation geometries. The strong gradient of water vapour in the atmosphere, always required an implicit knowledge of its vertical distribution often "hidden" in the way the conversion from slant to vertical column densities (SCD to VCD) via the calculation or estimation of the air-mass factor (AMF) has been approached.

The paper is an important contribution to this problem, since it approaches this issue for the first time explicitly, and shows convincing improvements, especially when retrieving TWVC in the vicinity of clouds, or evaluating, and improving the performance for various surface reflectance conditions. However the exact relation between cloud coverage, cloud height and retrieval performance remains to this reviewer still - at least to some extent – obscure and while I can highly recommend the paper for publication, I would like the authors to address this and a few other issues before.

1. Scale-height parameterization

The paper goes in depth on a specific parametrising of the a priori (better "first guess") water vapour profile using a parameterization of the water vapour scale height. While the motivation to introduce knowledge on the water vapour profile is in principle clear to any reader familiar with TWVC DOAS-like approaches, for the non-expert reader, the relation to cloud screening and surface sensitivity is not apparent. The simplest solution to use a "first-guess" water vapour profile from NWP re-analysis data is simply excluded with a reference to Wang. Then GPS climatologies are used to derive a scale-height parametrization. However, it is not made explicit or clear that the rational to use a scale-height (instead of the probably already quite realistic NWP full profile) is probably the simple need to regularize, and therefore constrain the retrieval problem. Since instruments measuring in the UV-visible range, will not be able to retrieve water vapour with significantly more than 1 to 2 independent pieces of information in the vertical. Adjusting VCD sub-columns by changing a single parameter, ie. scale height,

therefor serves the need to constrain the problem. Since otherwise using re-analysis data as a-priori or first guess (depending on the inversion approach) and adjusting multiple layers in the retrieval would clearly have been the better approach. Such a parameterization using a scaled full profile, then however, also has the tendency (or advantage) to compensate for missing information (e.g. below the cloud), which is essentially missing in the measurements. To which extend this happens here is not very clear, and leads to the next issues concerning the treatment of cloud-coverage.

2. Treatment of clouds

The treatment of partially (or fully) cloudy scenes (as observed and expressed in geometrical cloud coverage (cloud fraction CF) with the use of collocated imager data) is critical for TWVC retrievals, since clouds may shield or amplify (through scattering) the true total column value. Up to the validation section, (Section 2 to 5), the paper discusses the conversion of SCD to VCD for any level of cloud fractions, using the independent pixel approximation. The AMF error analysis however seems to be carried out for a CF of up to 50%. The impressive results shown in Figure 13, present the results for CF<20% and for all-sky (CF up to 100%), arguing for the usage of the presented scale-height method. These results seem to indicate that the method even works for CF>20%. In the introduction to the validation section (Section 6) it is then however stated that the validation is carried out for CF of up to 20% ("clear-sky") only and the paper validates the results with SSMI in dependence of CTH in Figure 21. It is assumed that this evaluation is also for CF<20%. Otherwise, the reader would expect significant underestimations of the results for large CF with high CTH. So Figure A10 to 13 adds to the confusion since, the results presented there seem to indicated the opposite: high cloud fractions for high cloud top levels lead to overestimations with respect to SSMI. How is this result interpreted with respect to the used combination of independent pixel approach, WV profile scale-height parameterization method, and the evaluation of the AMF and its error? What is eventually seen by the authors as the final product and at which CF? The CF threshold has been key to all previously

published retrieval methods from the UV/visible to NIR spectrometers. Therefore one would expect a clear statement even up-front in the introduction and for sure in the conclusions, if the product, with its novel scale-height approach, wants to target a specific cloud-coverage threshold or is proposing one for final use.

Minor comments:

l.57 p.2: Is the wavelength alignment carried out for all solar measurements provided by S5p, or only once for all retrievals.

l.73, p3: Are ISRF changes in width found with the WV retrieval over the orbit for S5p?

l. 84, p.3: the difference between ground and spectral pixel should be made clear to avoid confusion.

l.101, p.4: In this equation "beta" is not defined and it is not clear how I0 from the solar irradiance is used here.

Eq4, p.5: Is the refractivity equation relevant here? I guess the important point to be made is the use of COSMIC water vapour climatologies in contrast to the already smoothed model data. It is still very puzzling why the former should be better for this purpose than using actual reanalysis data. Since GPS (and hyper-spectral TIR) profile data is meanwhile an essential component in NWP data, and the model helps in reducing the vertical information content towards the one from S5p.

p. 4: The VCD equation on page 4 is not numbered and the usage of I is confusing here, since I guess it refers to the iteration step instead of the previously used sub-column layer number. Otherwise some clearer explanation would be needed.

---

## Author Response (AR1)

We revised the manuscript in line with the recommendations from the referees. In the following, we will list the most important modifications with respect to the AMTD version of the manuscript. A pdf manuscript compiled with LatexDiff is attached to this "Author's Response".

- We added further explanations of the advantages of the visible "blue" spectral range in the introduction.
- We also provide a brief introduction of the relation between the water vapour profile and cloud screening and surface sensitivity in Section 2.
- In Section 3 we give more information about the COSMIC water vapour profile data in a separate subsection and added few sentences about our procedure of the scale height parameterization at the beginning of the section.
  Also, we calculated AMF from ERA-5 water vapour profiles for the orbit including an atmospheric river and compared the resulting VCDs with the VCDs from the iterative scale height approach.
- We further included error estimations for ISRF biases in Section 5.
- To make the manuscript more concise, we rearranged the figures and moved former Figures 5,6, A3-A6, and A8-14 to the Supplementary Material.
- In the Appendix B we extended the discussion about the choice of the water vapour cross-section by adding plots of the respective absorption cross-section spectra.
- We carefully inspected each figure and modified the labels, legends, etc. if necessary.

We would like to thank the referee for the positive assessment and several helpful comments. Below we reply to the issues raised by the referee, where
blue repeats the reviewer's comments,
black is used for our reply,
*and green italics is used for modified text and new text added to the manuscript.*

Review of Total Column Water Vapour Retrieval from S-5P/TROPOMI in the Visible Blue Spectral Range by Christian Borger et al.

This paper presents a study where the authors have developed an algorithm to retrieve total column water vapour (TCWV) from 'blue' portion of the visible spectrum band on Sentinel 5-P/TROPOMI, and make comparisons with some validation sources.

This is an interesting study, and is a good example of exploiting the potential satellite data beyond that which was intended by the developers of TROPOMI. It is certainly of interest to the AMT community, and I recommend publication, if the following points and concerns are addressed.

We thank the reviewer for the general positive statement.

General Comments

In general, the paper lacks context to the wider TROPOMI picture, and only very briefly states the justification for why the authors attempt TCWV retrieval in the blue part of the visible spectrum. Even here, this discussion is in relation to the 'red' portion of the visible spectrum (where the authors have experience), and not products from other spectral regions (e.g. the SWIR). Especially since the errors can get quite high with these retrievals (up to 50%), I think the authors need to discuss the added benefit of retrievals from the 'blue' band to TROPOMI. The authors very briefly give some discussion to this in the conclusions (processing time, etc), this needs to be expanded upon. The authors identify TCWV products from other spectral regions on other instruments (e.g. TIR from AIRS), but there is no discussion on how TCWV from the 'blue' band compare to the TCWV products from these other instruments and spectral regions. This discussion needs to be included into the paper.

Section 1 – The Introduction.

Context is lacking in this section, it is unclear as to the advantages of retrieving TCWV from this particular spectral region. Given the other spectral bands and satellites that the authors mention, and not just the 'red portion' of the spectrum.

We added further explanations of the advantages of the visible "blue" compared to other spectral ranges and satellites. These include 1) similar sensitivity for ocean and land surface allowing for global coverage, 2) possible retrievals under partly-clouded conditions, 3) compared to the thermal IR a much higher sensitivity for the near-surface layers and 4) simple spectral analysis, i.e. no need for forward model calculations. The following text is added to the introduction:

*The visible spectral range is particularly interesting for the retrieval of total column water vapour (TCWV): in contrast to the microwave range it has a similar sensitivity for ocean and land surface allowing for global coverage. Also, it is possible to conduct retrievals under partly-clouded conditions and, in comparison to the thermal infrared, it has a much higher sensitivity for the near-surface layers. Furthermore, the spectral analysis is straightforward, i.e. no forward model calculations are necessary.*

Indeed, the authors have not referenced the study covering water vapour retrieval from the 2305-2385 nm TROPOMI spectral band (Schneider et al., 2020), which would have been in the AMT discussion forum at time of submission. They have also missed water vapour retrieval studies from the Shortwave Infrared (SWIR) in general, this is very strange since the SWIR gives good sensitivity to the surface e.g. (Trent et al., 2018), and surface sensitivity is clearly one of the key selling points for the 'blue' region.

Thanks for the hint of the missing reference, we added it in the revised manuscript. We had a look at the $H_2O$ data provided by Schneider et al. (2020) and compared them to our dataset for the arbitrary chosen time-range 2 to 4 December 2018. As TROPOMI's SWIR and the vis bands have different pixel sizes and collocations, we gridded the data to a 0.25°x0.25°. We only selected clear-sky observations, i.e. with cloud fractions < 20% in the visible spectral range. As Schneider et al. (2020) do not provide cloud fraction data, we only compare those gridcells that are classified valid for our TCWV data. For a better understanding of the differences, we separated the data into data over ocean and data over land. The results are given in the Figures below (left panel: visible "blue" TCWV; right panel: SWIR TCWV).

[Figure]

H2O VCD comparison, 2-4 Dec 2018, land

H2O VCD comparison, 2-4 Dec 2018, ocean

The "blue" TCWV shows excellent agreement to ERA-5 over both surface types with slopes around 1.0 and $R^2$ of 0.92 and 0.95.

In contrast, though the large majority of the SWIR TCWV data are concentrated along the 1-to-1 diagonal over land, we also find a large point cloud which negatively influences the linear fit. Over ocean, a similar point cloud can be observed, and the majority of the points are not concentrated along the 1-to-1 diagonal. Reasons for these major discrepancies could be the missing cloud information within the SWIR data and maybe also saturation effects of the retrieval. Additionally, the low sensitivity of the SWIR retrieval for near-surface layers over the dark ocean surface could lead to potential discrepancies.

**Section 3 – A priori water vapour**

While the aim of this section is clear, the story of how the methods that are used are a little unclear, and the general message gets a little lost in the details. Here in section 3, I think a few sentences that summarises the processes that are undertaken in sections 3.1 and 3.2 would be useful.

We added further explanations of our steps within the Section's introduction The following text is added:

*We proceed as follows: First, we evaluate how well the method used to calculate the water vapour scale height can reproduce the COSMIC profiles via an AMF comparison. Then we examine how the scale height can be parameterized globally and investigate for a parameterization over ocean and land separately. Finally, we implement the parameterization in an iterative retrieval scheme and evaluate the new estimates of the $H_2O$ VCD.*

We also rearranged the introduction and moved the description of the COSMIC data into a separate subsection.

Readers may well be unfamiliar with COSMIC and ROMSAF, and some background would be beneficial here. Given the importance of the a priori profile on the results, some discussion on the biases associated with using COSMIC as a base would be welcome.

We added further details on the used COSMIC data and brief discussion on possible biases associated with using COSMIC within the new subsection in Section 3:

*3.1 COSMIC water vapour profiles*

*For our investigations we use profile data retrieved from measurements of the Constellation Observing System for Meteorology, Ionosphere, and Climate (COSMIC, Anthes et al., 2008) program provided by the Radio Occultation Meteorology Satellite Application Facility (ROMSAF). The COSMIC data are based on the GPS radio occultation (RO) technique, which provides high resolution vertical profiles of bending angles (Hajj et al., 2002) that can be used to retrieve the atmospheric refractivity. Since the atmospheric refractivity is dependent on the air pressure, the air temperature, and the water vapor pressure (Smith and Weinstraub, 1953), GPS RO allows for the retrieval of profile information under all-weather conditions with a high vertical resolution of approximately 100m in the lower troposphere up to 1km in the stratosphere (Anthes et al., 2011) and an accuracy of around 1g/kg (Heise et al., 2006; Ho et al., 2010b) while having an almost uniform global distribution (Ho et al., 2010a).*

*The ROMSAF profiles have been retrieved via a 1D-VAR scheme within a reprocessing initiative for creating climate data record (CDR) v1.0. Given the strict product requirements and the validation studies with ERA-Interim and radiosondes (Nielsen et al., 2018), biases associated with using COSMIC should be of secondary order.*

*We use data retrieved between 2013 and 2016, which accumulates to approximately $1.6 \times 10^6$ profiles.*

Section 6 – Validation study.

More detail about the SSMIS and SuomiNet data are needed here, fundamentally why did the authors choose these measurements for their validation, and why are they appropriate for TROPOMI inter-comparisons?

SSMIS and SuomiNet measure in the microwave and radio spectral range, respectively, which allows for the retrieval of TCWV under all-sky conditions with high accuracy (~1kg/m$^2$ and ~2kg/m$^2$, respectively). Thus both datasets are widely considered as gold standard and used as references within our study. These informations are given in the introduction of every subsection of the validation section.

In the introduction the authors identify some specific advantages of retrieving TCWV from the blue part of the spectrum as opposed to the red, which has been done previously. To me, the validation section would have been a good opportunity to compare against TCWV data from the red part of the spectrum, not necessarily from TROPOMI, but from other satellites. Indeed comparisons from other 'blue' TCWV measurements from OMI as identified by the authors would have been useful, to identify any potential differences between the spectral regions. Or, please justify their exclusion from the study. Validation is largely done using TCWV retrieved from microwave instruments, are there any particular biases associated with TCWV retrievals, when compared against the visible band?

There a lot of TCWV datasets available that could be used for validation studies. We decided to restrict ourselves to SSMIS and SuomiNet, because both data sets are based on measurements which are insensitive to clouds and are known for their proven high accuracy. In the case of OMI, it has to be taken into account that the instrument is affected by a row anomaly since 2007 and hence almost half of the data have to be used with caution. Also, OMI does not cover the red spectral range. Nevertheless we had a look at the OMI TCWV data provided by Wang et al. (2019). We applied the required filter criteria (cloud fraction < 5%, RMS < 0.001, MDQFL=0, cloud pressure > 750hPa, and TCWV < 75kg/m$^2$) and also destriped the OMI $H_2O$ SCD. For our TROPOMI data we only applied the proposed cloud fraction filter. As comparison time range we selected December 2018. Since OMI has a much larger pixel size than TROPOMI, we gridded both data to 0.25°x0.25° grid. For a better understanding of the differences, we separated the data into data over ocean and data over land. The Figures below show the results of the comparison.

[Figure]

The 2D histograms reveal that for both surface types the products do not agree well: the OMI VCDs are much larger than our TROPOMI VCD, by almost 30%. However, considering the higher signal-to-noise ratio of TROPOMI and the findings of our validation study using even looser filter criteria than Wang et al. (2019), we conclude that the TROPOMI VCDs are more reliable than the OMI VCDs.

**Other**

With regards to the English, the paper could be improved given a review by a native English speaker. I will not go into detailed specifics in this review, but this paper would benefit from a greater use of punctuation (i.e. commas).

We agree that our English (grammar, spelling, punctuation) is not perfect as we are no native speakers. We carefully checked the manuscript with respect to English language and punctuation. In addition, Copernicus will provide a thorough language editing before final publication.

There are a large number of figures in this paper, and while I applaud the efforts of the authors for the detail they have gone into, 26 figures + 14 in the appendices is unneeded for a journal paper, which should only be showing the key highlights. I suggest to the authors to place some of these figures in supplementary materials.

We rearranged our figures in the revise manuscript. We have moved former Figures 5,6, A3-A6, and A8-14 to the Supplementary Material.

In addition, on some of the Figures the axis labels and legends are quite small (e.g. Figure 5), I recommend that the authors increase the font size.

Thanks for the hint. We carefully inspected every figure and modified the labels, legends, etc. if necessary.

**Additional thoughts**

I note that the authors have stated that their dataset is available on request, I would strongly encourage them to place their dataset into an online repository.

Currently, we are working on the improvement of the input parameters of the retrieval, e.g. we are working on a standalone cloud fraction and cloud top height product. Both are major tasks and will take several months to be available. Thus we cannot include the improved cloud products in the current study. However, in light of the expected improvements to come, we consider the presented TCWV product not as the final version and refrain from an explicit publication of this dataset But we intend to provide the updated data set after these improvements have been implemented. The current data will be made available on request.

**Specific comments.**

On p3, line 75. The authors state that the absorption is weak in the fit window, and hence the line lists vary. I am not convinced by this argument, to me just because absorption is weak the uncertainty of the spectral lines shouldn't necessarily be higher, does HITRAN state this? What exactly is different between the databases, is it the position of the lines, the number of lines?

A similar question has also been risen in the Short Comment by Eamon Conway. What we meant by "variation" of the line list is the distinctive disagreement/change of the strength of the peak absorption between the different HITRAN versions. Please see the figure below: The left panel depicts the high-resolution cross-sections and the right panel the same cross-section convolved with the TROPOMI ISRF. Though the high-resolution cross-sections seem to be quite similar, the convolved cross-sections reveal an alternating pattern going from HITRAN 2008 to 2016.

[Figure]

We added the figure above and the following text to the appendix section of the water vapour cross-section:

*Figure A3 compares the absorption cross-sections of the different HITRAN versions. For the high-resolved cross-section (left panel) the differences between the versions are hardly visible, however, after the convolution with the TROPOMI ISRF (right*

*panel), distinctive differences in the peak absorption are clearly visible: in comparison to HITRAN2008, the absorption peak of HITRAN2012 is approximately 7-9% higher than HITRAN2008 and the absorption peak of HITRAN2016 is approximately 7-9% lower than HITRAN2008.*

P4, line 91. Why are the data shown from Figures 1 and 2 based on different orbits? It'd be more consistent to show the same orbit results.

We changed Figure 1 accordingly and took an example from the orbit depicted in Figure 2.

P5, Equation (4) – The calculation of atmospheric refractivity is highly dependent of air pressure and air temperature. Therefore the scale height for the water vapour profiles is highly dependent on knowledge of these factors. How accurate is this knowledge, and how sensitive is the calculation of scale height to errors in the knowledge of air pressure and air temperature?

ROMSAF requests in their product requirements a target pressure accuracy less than 1 hPa from 0-50km and a target temperature accuracy of 1-2K from 0-5km and 1K from 5-30km. The provided geopotential height thus has an accuracy of around 50-500gpm in the troposphere which is approximately 5-50m in geometric height (increasing with altitude). Considering typical specific humidity errors of around 1g/kg, errors in the altitude grid are negligible within the scale height calculations.

P5, line 143. Why 63% (assuming this is total water vapour up to 150 hPa)? Is this because there is not much water vapour above this point? If so, this should be stated.

63% is the fraction of total vertical column that should be encountered within the sub-column between ground and first scale height: $1-1/e = 0.63 \rightarrow 63\%$

P5, line 147. Why 7%?

7% is a typical value for the ocean surface albedo (Tilstra et al., 2017).

P6, I think Figures 5 and 6 could be moved to the appendices or supplementary material. For Figure 5, it is very unclear to me exactly what causes the 'bad' profiles, as opposed to the good profiles, this needs to be expanded upon.

We have moved Figure 5 to the Supplementary Material. The "bad" profiles occur if a sharp decrease of the water vapour concentration with altitude exists. Such profiles occur when a moist boundary layer is topped by a dry free atmosphere. The "good" profiles are associated with a well-mixed troposphere and thus decrease with altitude

following an exponential decay. This is clarified in Section 3.2 of the revised manuscript as follows:

*In general, bad agreement (left column) occurs for profile shapes in which a sharp gradient is observed in the lower troposphere and from that quasi-constant values with altitude. Such profiles usually occur when a moist boundary layer is topped by a dry free atmosphere. Nevertheless the maximal absolute relative AMF-deviations only have values around 15%. In contrast, good agreement (right column) is found for profile shapes following an exponential decay with altitude, which indicates a well-mixed troposphere.*

P7, line 187. How are ocean and land differentiated? What about heterogeneous scenes or lakes?

We use a land-sea mask derived from GSHHS coastline data, in which we use the pixel center coordinates for the separation into land and ocean. As the NDVI is not available over lakes, we treat them as ocean.

Figure 14. The authors claim that there is a distinct separation of the H2O VCD between land and ocean, which vary between albedo versions. I did see it eventually, but it is quite subtle, this might be represented better if focused upon? Also, I don't feel that the inclusion of this Figure is particular beneficial to the paper, and can be placed in supplementary material.

We changed Figure 14 and highlighted the areas of distinct separation between land and ocean with black circles in each subfigure.

P10, section 5. There is no discussion on instrumentation errors such as ILSF biases and radiometric errors. Can you comment on the impact of these?

Radiometric errors are already included within the fit error of the DOAS analysis. Considering ISRF/ILSF biases we investigated the impact of using a Gaussian ISRF instead of an asymmetric Super-Gaussian and compared the resulting $H_2O$ SCDs (see Figure below) for the same orbit as in Figure 2 of our paper.

[Figure]

In principle the SCDs using the Gaussian ISRF are slightly higher (by about 1%) than the SCDs using the asym. Super-Gaussian ISRF. We added this Figure to the revised manuscript and also added the following text to Section 5.1:

*To estimate errors associated with ISRF biases, we calculated the $H_2O$ SCD using a Gaussian ISRF (instead of an asymmetric Super-Gaussian) for orbit 6930 and compared them to the SCDs from the "standard" retrieval setup for a SZA < 88°. The comparison depicted in Figure S3 reveals that the SCDs using the Gaussian ISRF highly correlate with the "standard" SCDs and only differ by approximately 1%. Considering the much higher fit errors, errors due to biases in the ISRF are negligible.*

P11, line 316. Is the spectroscopic uncertainty not as significant as any of these other factors? It would be useful to state the impact of this uncertainty in relation to the others. These are mentioned in the summary, but would be useful in the main text as well.

The statement about errors of spectroscopic uncertainty is now included in the description of "Uncertainties in the slant column density" (Section 5.1.):

*Considering the LP-DOAS comparisons (see Sect. 2.1 and Appendix B) we estimate these errors to be around 5% for this study.*

P15, summary and conclusion. Can you comment on the uncertainty of the retrievals (you say up to 50%) in comparison to TCWV retrievals from other spectral bands.

The uncertainty of our retrieval is around 10-20% for favourable and 20-50% for unfavourable observation conditions. Most uncertainty estimates of retrievals in other

studies are based on validation studies rather than theoretical calculations and thus only provide absolute values of the error. For example Bennartz and Fischer (2001) estimated an error of 2.5kg/m$^2$ for their MERIS TCWV product, but most of their reference measurements were taken under dry atmospheric conditions of 20-25kg/m$^2$ (Bauer 2009). Hence (and taking into account our additional comparison studies) we conclude that our estimated uncertainty is within or even better than the typical range of NIR and IR TCWV retrievals.

P15, lines 469-471. Here the authors state that the retrieval allows for a fast execution of large datasets. This to me is one of the key benefits of the retrievals in this waveband, however this is the first time that it has been mentioned in the manuscript. A brief discussion on processing times, and comparisons with other TCWV products would be beneficial.

As we are using a linearized retrieval scheme, the spectral analysis of one TROPOMI orbit takes approximately 1 min (depending on the IT system). As a rule of thumb Beirle et al. (2013) found an increase of speed of 3 orders of magnitude by going from non-linear to linear fit for their MATLAB routine (see Table 3 in their paper). We added the following text to Section 2.1:

*According to Beirle et al. (2013) the computational speed increases by 3 orders of magnitude by going from non-linear to linear fit for their MATLAB routine (see Table 3 in their paper).*

Technical P2, line 45 – A reference should be added for the increase in TROPOMI spatial resolution to 3.5x5.6.

We added a reference to the TROPOMI L1b Product Readme File (Rozemeijer and Kleipool, 2019).

P3, line 83 – Band 4 is mentioned for the first time, please identify what TROPOMI Band 4 is.

We provided further information on Band 4. We changed the text in Section 2.1 accordingly:

*Due to the high daily data volume of the TROPOMI L1B radiances the execution of a non-linear fit without high performance infrastructure is demanding in computation time. For instance TROPOMI's UVIS Band 4, which covers the spectral range of 400-499nm, generates about 40 gigabyte per day. Therefore, we implemented a weighted linear least squares fit …*

A number of the equations appear to be missing equation numbers, e.g. the AMD calculations p4, lines 106, 111.

We carefully revised the numbering of the equations.

P5, line 120 – Could you provide an explicit statement for what is meant by scale height please.

We assume a water vapour profile following an exponential decay with altitude, i.e. $\rho_w(z) \sim \exp(-z/H)$ where H is the scale height of water vapour. Thus, we define H as the altitude (from ground) at which $1-1/e=63\%$ of the TCWV are accumulated. The text in the introduction in Section 3 is now:

*Weaver and Ramanathan (1995) approximated the water vapour profile by an exponential decay with altitude:*

*$n_v(z) = n_0 e^{-z/H_v}$*

*where $H_v$ is the scale height of water vapour, which they defined as: …*

P5, line 130 – Please define ROMSAF.

We added the full name of the acronym.

P5, line 135 – GPS RO allows to retrieve profile information -> GPS RO allows for the retrieval of profile information

Thanks for this hint.

P15, line 468 – and Sentinel 5

We also added Sentinel 5.

P19, line 534 – greak -> great

Thanks for the hint.

We would like to thank the referee Rüdiger Lang for the constructive and helpful comments and questions. Below we reply to the issues raised by the referee, where
blue repeats the reviewer's comments,
and black is used for our reply,
*and green italics is used for modified text or text added to the manuscript.*

The paper by Borger et al., on total column water vapour retrievals from Sentinel-5P (S5P) is demonstrating the large potential of water vapour retrievals in the blue and visible spectral range to yield an accurate estimate of the total water vapour column(TWVC) largely independent from model data and capable to cover all surfaces. This type of TWVC product from instrumentation like GOME-1/2, SCIAMACHY, OMI and TropOMI, therefore serves as an important product for the evaluation of (re-)analysis NWP model data output.

The paper by Borger et al. is overall well written and structured and apart from presenting the very first results of this type of TWVC product from the S5p mission the paper also presents an interesting and novel approach of employing sub-column water vapour profile information to TWVC retrievals, via a parameterization of the water vapour atmospheric scale-height for various conditions, like surface type and different observation geometries. The strong gradient of water vapour in the atmosphere, always required an implicit knowledge of its vertical distribution often "hidden" in the way the conversion from slant to vertical column densities (SCD to VCD) via the calculation or estimation of the air-mass factor (AMF) has been approached.

The paper is an important contribution to this problem, since it approaches this issue for the first time explicitly, and shows convincing improvements, especially when retrieving TWVC in the vicinity of clouds, or evaluating, and improving the performance for various surface reflectance conditions. However the exact relation between cloud coverage, cloud height and retrieval performance remains to this reviewer still - at least to some extent – obscure and while I can highly recommend the paper for publication, I would like the authors to address this and a few other issues before.

We thank the reviewer for his positive general statement.

1. Scale-height parameterization

The paper goes in depth on a specific parametrising of the a priori (better "first guess") water vapour profile using a parameterization of the water vapour scale height. While the motivation to introduce knowledge on the water vapour profile is in principle clear to any reader familiar with TWVC DOAS-like approaches, for the non-expert reader, the relation to cloud screening and surface sensitivity is not apparent.

We thank the reviewer for this note and added further explanations on the relation between the water vapour profile and cloud screening and surface sensitivity in Section 2.2 in the revised manuscript. The following text and Figure have been added to the revised manuscript:

[Figure]

*Figure 3 depicts typical examples of BAMF profiles for different clear- and cloudy-sky scenarios. The AMFs for the cloudy-sky scenarios were calculated assuming a surface albedo of 7% and an effective cloud fraction of 20%. For the clear-sky scenario (left panel) the sensitivity decreases towards the surface. For the cloudy-sky scenarios (right panel) the BAMF profiles slightly increase towards the (bright) cloud top surface of the respective scenario. Below the cloud, the sensitivity is 0, because the atmosphere is shielded. Since high clouds shield large fractions of the atmosphere and hence also of the water vapour column below the cloud (see black dashed curve), the AMF has to be corrected correspondingly and thus decreases for increasing cloud top heights.*

The simplest solution to use a "first-guess" water vapour profile from NWP re-analysis data is simply excluded with a reference to Wang. Then GPS climatologies are used to derive a scale-height parametrization. However, it is not made explicit or clear that the rational to use a scale-height (instead of the probably already quite realistic NWP full profile) is probably the simple need to regularize, and therefore constrain the retrieval problem. Since instruments measuring in the UV-visible range, will not be able to retrieve water vapour with significantly more than 1 to 2 independent pieces of information in the vertical. Adjusting VCD sub-columns by changing a single parameter, ie. scale height, therefor serves the need to constrain the problem. Since otherwise using re-analysis data as a-priori or first guess (depending on the inversion approach) and adjusting multiple layers in the retrieval would clearly have been the better approach. Such a parameterization using a scaled full profile, then however, also has the tendency (or advantage) to compensate for missing information (e.g. below the cloud), which is essentially missing in the measurements. To which extend this happens here is not very clear, and leads to the next issues concerning the treatment of cloud-coverage.

We agree with the reviewer that the simplest solution for an a priori water vapour profile is the usage of NWP profiles. Nevertheless, our goal was to be independent from any model data, as there are some issues to consider when using profiles from

NWP or reanalysis: Models could be affected by a systematic bias in their simulations and, for the case of reanalysis, are also affected by the varying numbers of observations. Furthermore, current global models typically have a spatial resolution that is significantly larger than a TROPOMI pixel ($3.6 \times 5.6 km^2$ vs $25 \times 25 km^2$). In addition, the temporal resolution is also limited. As a result, the model is unable to adequately reproduce sub-scale/sub-grid processes (e.g. cloud cover/height). This also raises the question to what extent the modelled atmosphere is trustworthy, e.g. if the modelled WV profile does not coincide with observations.

[Figure]

To illustrate this conflict of potentially wrong WV profiles, we calculated the AMF using water vapor profiles from the reanalysis model ERA5 (hourly timestep, $0.25 \times 0.25°$ grid, 60 vertical levels) for the same orbit as in Figure 13 and compared the resulting VCDs. It can be seen that, especially in cloudy areas, the VCDs from ERA5-AMFs clearly overestimate compared to the VCDs from iterative scale height AMF (e.g. in the area of the Atmospheric River by 30°N).

[Figure]

Further analyses of the water vapour profiles in these particular regions reveal that ERA5 underestimates the VCD above the cloud in comparison to the iterative scale-height (ISH) method. In addition, the variation of the above-cloud mean WV profiles of ERA5 is much smaller than those of the ISH method which could indicate that for these cases ERA5 tends more to its a priori information, i.e. climatological mean.

Nevertheless further investigations are beyond the scope of this paper and should be addressed in later studies.

We included the VCD from the ERA5 profiles as well as the figure of the profiles in the revised manuscript and added the following text to Section 3.4:

*Taking a closer a look at the reasons for the deviations of results retrieved for the ERA-5 profiles, Fig. 13 depicts the mean of the normalized water vapour profiles of ERA-5 and the iterative scale height approach for the AR region (around 30°N). The left panel of Fig. 13 shows the water vapour profile from ground up to 15km. In comparison to the iterative approach, ERA-5 is much drier above approximately 2.5km for these particular cases, indicating that ERA-5 might systematically underestimate the water vapour content above the cloud within the region of the atmospheric river. This finding is further supported by the right panel of Fig.13 which illustrates the normalized water vapour profiles above the cloud top: ERA-5 profiles are close to 0 and show only small variations, whereas the profiles of the iterative approach indicate higher water vapour concentrations along with a much higher variability. One potential reason for the discrepancies of ERA-5 could be the missing of observational input data for the reanalysis: without observations, the reanalysis model is dominated by its a priori information (e.g. a climatological mean), so that it can be systematically distorted from the real atmosphere. However, further investigations of possible ERA-5 biases are beyond the scope of this paper.*

**2. Treatment of clouds**

The treatment of partially (or fully) cloudy scenes (as observed and expressed in geometrical cloud coverage (cloud fraction CF) with the use of collocated imager data) is critical for TWVC retrievals, since clouds may shield or amplify (through scattering) the true total column value. Up to the validation section, (Section 2 to 5), the paper discusses the conversion of SCD to VCD for any level of cloud fractions, using the independent pixel approximation. The AMF error analysis however seems to be carried out for a CF of up to 50%. The impressive results shown in Figure 13, present the results for CF<20% and for all-sky (CF up to 100%), arguing for the usage of the presented scale-height method. These results seem to indicate that the method even works for CF>20%. In the introduction to the validation section (Section 6) it is then however stated that the validation is carried out for CF of up to 20% ("clear-sky") only and the paper validates the results with SSMI in dependence of CTH in Figure 21. It is assumed that this evaluation is also for CF<20%. Otherwise, the reader would expect significant underestimations of the results for large CF with high CTH. So Figure A10 to 13 adds to the confusion since, the results presented there seem to indicated the opposite: high cloud fractions for high cloud top levels lead to overestimations with respect to SSMI.

In Figures 20, 21, 23, 24, and 26 we investigated the CTH dependence for CF<20%, which would correspond to the first two columns of Figures A10 to 13. To avoid confusion we added further explanations in the captions of Figures 20, 21, 23, 24, and 26. We added the following phrase to the Section's introduction:

*For the sake of completeness, we also briefly investigate higher cloud fractions at the end of each subsection and provide the results in the Supplementary Material.*

How is this result interpreted with respect to the used combination of independent pixel approach, WV profile scale-height parameterization method, and the evaluation of the AMF and its error?

In very special cases problems with using the IPA can occur due to TROPOMI's small pixel size. However, in general the same effects also occur in cloud retrieval and in cloud correction of the AMF. These two effects largely compensate each other.

What is eventually seen by the authors as the final product and at which CF? The CF threshold has been key to all previously published retrieval methods from the UV/visible to NIR spectrometers. Therefore one would expect a clear statement even up-front in the introduction and for sure in the conclusions, if the product, with its novel scale-height approach, wants to target a specific cloud-coverage threshold or is proposing one for final use.

In fact, the results imply a possible use of the data up to cloud fractions of 100%, but it must be taken into account that for high cloud fractions no information below the cloud can be gained. Also the input parameters (clouds & albedo) are currently still subject to relatively large uncertainties and are continuously improved. For example,

the OMI albedo is still used for the calculation of the cloud fraction, so that if there will be an update to an improved albedo data set derived from TROPOMI, significant changes in the cloud fraction can be expected. Such changes in CF also lead to changes of the cloud height. Furthermore various specific cloud height updates are also being implemented at the moment. Due to the continuous changes/improvements of the input parameters we refer the expert user to the results of the error estimation depicted in Figures 17 and 18 and summarized in Table 4 and 6. We also added a clear statement in the revised manuscript and recommend the non-expert user to only use VCDs with CF<20% and AMF>0.1, which represents a good compromise between coverage and retrieval accuracy. The following can be found now in the summary and abstract:

*For the general purpose we recommend to only use VCDs with cloud fraction < 20% and AMF > 0.1, which represents a good compromise between spatial coverage and retrieval accuracy.*

Minor comments:

l.57 p.2: Is the wavelength alignment carried out for all solar measurements provided by S5p, or only once for all retrievals.

The wavelength calibration is performed for each day with the latest available daily irradiance. We added this information to Section 2.

l.73, p3: Are ISRF changes in width found with the WV retrieval over the orbit for S5p?

We do not find significant ISRF changes along the orbit as reported by Beirle et al. (2017) for the GOME-2 instrument which is probably due to the TROPOMI's better cooling system compared to GOME-2. Nevertheless we see changes of the ISRF width over cloudy scenes which might indicate differences in the pixel illumination.

l. 84, p.3: the difference between ground and spectral pixel should be made clear to avoid confusion.

This point was made clearer in the updated version.

l.101, p.4: In this equation "beta" is not defined and it is not clear how I0 from the solar irradiance is used here.

The respective explanations were added. To avoid confusion with other sections, we changed the variable names of beta to mu and from $VCD_i$ to $c_k$. The text is now:

*These simulations yield a Jacobian vector $J=d\ln I / d\mu$ (with the absorption coefficient $\mu$ and the simulated intensity $I$ at TOA normalised by the solar spectrum $I_0$) defined at each grid box $k$.*

*These BAMF profiles have to be combined with the partial vertical columns $c_k$ of an a priori water vapour profile: …*

Eq4, p.5: Is the refractivity equation relevant here? I guess the important point to be made is the use of COSMIC water vapour climatologies in contrast to the already smoothed model data. It is still very puzzling why the former should be better for this purpose than using actual reanalysis data. Since GPS (and hyper-spectral TIR) profile data is meanwhile an essential component in NWP data, and the model helps in reducing the vertical information content towards the one from S5p.

As Reviewer #1 also states that this equation only creates confusion, we removed it in the revision process.

p. 4: The VCD equation on page 4 is not numbered and the usage of I is confusing here, since I guess it refers to the iteration step instead of the previously used sub-column layer number. Otherwise some clearer explanation would be needed.

We numbered all equations and modified them to avoid for potential conflicts between the different sections of the paper.

Literature

Beirle, S., Lampel, J., Lerot, C., Sihler, H., and Wagner, T.: Parameterizing the instrumental spectral response function and its changes by a super-Gaussian and its derivatives, Atmos. Meas. Tech., 10, 581–598, https://doi.org/10.5194/amt-10-581-2017, 2017.

Dear Eamon Conway,

Please find below our replies to the raised issues, where
blue repeats your comments,
and black is used for our reply.

We have a couple of comments on the way different editions of the HITRAN database are represented in this paper.

1. Regarding statement in Line 75. "The molecular absorption by water vapour within our fit window is relatively weak and hence the modelled line lists vary strongly from HITRAN 2008 to HITRAN 2012 (Rothman et al., 2013) and to HITRAN 2016 (Gordon et al., 2017)."

Please see attached figure, which shows the absolute absorption cross-sections ofHITRAN2008 and HITRAN2012 plotted on a linear scale, all isotopologues included. The cross-sections were all calculated using the HITRAN API with a temperature of 288K, 1 atm, 0.001 cm$^{-1}$ resolution and a 50 cm$^{-1}$ wing. The data sets appear similar and do not seem to "vary strongly". The residuals at 446.2 nm, 444.2 nm and 444.5 nm are due to improvements in calculated line positions in HITRAN2012. The line centers are slightly shifted, but they are present in each set. The HITRAN2012 line list is also more complete than HITRAN2008, note the extra HITRAN2012 weak absorption around 440 nm. Can the authors provide any evidence that suggest the data sets "vary strongly" in 430-450 nm?

We agree that looking at the high-resolved cross-sections, the differences between the HITRAN versions are small. However, for atmospheric remote sensing applications the high-resolved cross-sections have to be convolved with the instrumental spectral response function (ISRF) of the instrument. The figure below depicts the high-resolved $H_2O$ cross sections of HITRAN 2008, 2012, and 2016 (all at 296K) in the left panel and the corresponding ("low-resolution") cross-sections convolved with a typical ISRF of TROPOMI in the right panel. For the ISRF we assumed a symmetric Super-Gaussian with parameters taken from Beirle et al. (2017).

[Figure]

After the convolution one observes that the strengths of peaks of the cross-sections differ distinctively: HITRAN2012 is 7-9% higher and HITRAN2016 is 7-9% lower than

HITRAN2008. Ergo we find an alternating pattern between the cross-section versions, which is why we conclude that the versions "vary systematically" among themselves. In the revised manuscript, we added the figure to better clarify our view.

2. Regarding statement in Line 77. "Lampel et al. (2015) found out that HITRAN2012 underestimates the water vapour concentration derived from Long Path DOAS observations by approximately 10% and that the previous version HITRAN2008 agrees better to the reference measurements."

Within the discussion section in Lampel et al. (2015b), they found residuals in all their windows to be reduced by going from HITRAN2009 to HITEMP2010/HITRAN2012 cross-sections. (HITRAN2008 is HITRAN2009: the article was online in 2009 but the edition of the database is HITRAN2008.) Lampel et al. (2015b) also say "development of water vapour absorption compilations from HITRAN 2009 to HITEMP/HITRAN 2012 results in a better fit of the measurement data". This is not in line with what is claimed here? Can the authors please verify?

In our above mentioned statement we referred to the comparisons of LP-DOAS measurements to meteorological data made by Lampel et al. (2015). Their findings are summarized in Table 8 of their paper which gives regression results. The regressions yield a slope of 1.004 for HITRAN2008 and 0.918 for HITEMP/HITRAN2012, i.e. HITRAN2008 agrees better to the meteorological observations than HITRAN2012.

We did not discuss the fit quality and it is true that Lampel et al. (2015) found residuals decreasing by switching from HITRAN2008 to 2012. However, looking at the relative fit errors of the fit window around the strongest line (see Table 6 in their paper, W3) the error decreases from 1.04% to 0.98% for LP-DOAS.

Thus, in view of the observational evidence with almost negligible changes in the relative fit error, we see our statement confirmed.

3. Appendix B. The authors state that Lampel et al. (2015b) found the HITRAN2012 cross-sections to underestimate water vapor mixing ratios by 8% in 430-450 nm, while for HITRAN2008, the results are in excellent agreement with the meteorological station. The meteorological station is quoted by Lampel et al. (2015b) to have a 5% uncertainty on their value of humidity and a 2% error on the temperature. Lampel et al. (2015b) then proceeds to state "absolute differences of the cross-sections shown in Table 3 cannot be absolutely validated with sufficient precision". It therefore may not be true that HITRAN2008 is 'superior' to HITRAN2012 and 8% deviation is very close to the uncertainty of the measurement. Does this provide sufficient evidence to support the use of HITRAN2008 over HITRAN2012?

In Appendix B in our paper we present further LP-DOAS measurements conducted by Johannes Lampel and Stefan Schmitt at the CESAR Tower in Cabauw (Netherlands) during the CINDI-2 campaign. The $H_2O$ SCDs have been derived using HITRAN2012 cross-sections and the resulting water vapour volume mixing ratios have been compared to the meteorological measurements at different altitudes of the tower (see Figure A2 in our paper). From these further measurements we find underestimations

of 17% during day and 11% during night. These additional observations, combined with the findings of Lampel et al. (2015), provide sufficient evidence to use HITRAN2008 over HITRAN2012.

Literature

[revised manuscript text omitted]

**Figure S10.** 2D histograms for the comparison between TROPOMI and SSMIS f16 for July 2018 for different cloud fraction bins (left to right column) and cloud top height bins (top to bottom row). The color indicates the amount of points within one bin of each panel. The black dotted line indicates the 1-to-1 diagonal and the red solid line represents the results of the linear regression. The parameters of the linear regression and the coefficient of determination are given in the box in each panel.

[Figure]

**Figure S11.** Same as Fig. S10, but for SSMIS f17.

[Figure]

**Figure S12.** Same as Fig. S10, but for ERA-5 TCWV data over ocean.

[Figure]

**Figure S13.** Same as Fig. S10, but for ERA-5 TCWV data over land.

[Figure]

**Figure S14.** Scatterplots for the comparison between TROPOMI and SuomiNet for boreal summer 2018 for different cloud fraction bins (left to right column) and cloud top height bins (top to bottom row). The black dashed line indicates the 1-to-1 diagonal and the orange solid line represents the results of the robust regression. The parameters of the regression and the correlation coefficient are given in the box in each panel.